# An Updated Model of the Divisome: Regulation of the Septal Peptidoglycan Synthesis Machinery by the Divisome

**DOI:** 10.3390/ijms23073537

**Published:** 2022-03-24

**Authors:** Mohamed Attaibi, Tanneke den Blaauwen

**Affiliations:** Bacterial Cell Biology and Physiology, Swammerdam Institute for Life Science, University of Amsterdam, 1098 XH Amsterdam, The Netherlands; moattaibi@gmail.com

**Keywords:** divisome, FtsEX, FtsBLQ, FtsWI, FtsN, FtsZ, interactions, peptidoglycan hydrolases, septal peptidoglycan, regulation

## Abstract

The synthesis of a peptidoglycan septum is a fundamental part of bacterial fission and is driven by a multiprotein dynamic complex called the divisome. FtsW and FtsI are essential proteins that synthesize the peptidoglycan septum and are controlled by the regulatory FtsBLQ subcomplex and the activator FtsN. However, their mode of regulation has not yet been uncovered in detail. Understanding this process in detail may enable the development of new compounds to combat the rise in antibiotic resistance. In this review, recent data on the regulation of septal peptidoglycan synthesis is summarized and discussed. Based on structural models and the collected data, multiple putative interactions within FtsWI and with regulators are uncovered. This elaborates on and supports an earlier proposed model that describes active and inactive conformations of the septal peptidoglycan synthesis complex that are stabilized by these interactions. Furthermore, a new model on the spatial organization of the newly synthesized peptidoglycan and the synthesis complex is presented. Overall, the updated model proposes a balance between several allosteric interactions that determine the state of septal peptidoglycan synthesis.

## 1. Introduction

The divisome is a multiprotein dynamic complex driving bacterial cell division. The best known division mechanism is binary fission, for which the following processes need to proceed: chromosomal DNA replication and segregation, constriction of the inner membrane (IM), septal peptidoglycan (sPG) synthesis, and invagination of the outer membrane (OM) in Gram-negative bacteria [1]. The divisome regulates the latter three of these processes and is involved in the separation of genomes through the DNA translocase FtsK [2]. Divisome assembly at the division site is divided into two distinct stages separated by a significant time delay [3,4]. Early proteins include the tubulin-homologue FtsZ and its membrane anchors FtsA and ZipA, which form the proto-ring (Figure 1), together with several Z-ring-associated proteins (ZapA-G). The semi-essential ABC transporter FtsEX follows the ‘early’ proteins and is likely involved in further assembly of the divisome [5,6]. The earliest recruited ‘late’ protein is the DNA translocase FtsK, which is followed by a hierarchal recruitment of at least FtsQ, FtsBL, FtsWI and FtsN [3].

Division is a costly process for the bacterial cell, as it needs to increase in mass, double its genome, and produce a PG septum, which will be significantly degraded during daughter cell separation [7,8]. These processes rely heavily on the availability of sufficient nutrients, as is indicated by the close relationship between nutrient availability, bacterial metabolism, and cell cycle progression [9]. Division is determined for a complex regulatory web where the sum of multiple factors results in the start or delay of septum formation. Fortunately, the multifaceted regulation provides multiple angles for the development of new antibiotic compounds. In the context of the divisome, more than 20 inhibitors targeting the FtsZ protein have been discovered in the last decade alone [10]. However, historically speaking, compounds directly targeting peptidoglycan biosynthesis have been one of the most effective classes of antibiotics [11]. These include the most widely used β-lactam antibiotics that target PG synthesizing proteins, but also the glycopeptide antibiotics that target immature PG [11,12]. Thus, fruitful targets would be proteins directly involved in the (activation of) septal peptidoglycan synthesis.

The most significant and high-risk process during bacterial cell division is the synthesis and remodeling of the peptidoglycan layer at the division site. The periplasmic peptidoglycan is a large crosslinked polymer that surrounds the inner membrane and provides the cell with structural integrity and counterbalance to turgor pressure [13]. The layer, consisting of crosslinked strands of alternating *N*-acetylglucosamine (NAG) and *N*-acetylmuramic acid (NAM) residues, is of utmost importance for the structural integrity of the bacterial cell. However, peptidoglycan also has an additional role during bacterial fission, forming a physical septum between two future daughter cells [14]. Thus, it is no surprise that the bulk of the divisome is involved in peptidoglycan biosynthesis and remodeling.

Essential proteins directly synthesizing septal peptidoglycan are the glycosyltransferase FtsW and the transpeptidase FtsI (also known as PBP3) [15]. The activity and direct recruitment of FtsW and FtsI is regulated by a subcomplex consisting of FtsB, FtsL, and FtsQ [16]. FtsN is another regulatory protein, whose arrival at midcell leads to the activation of septal peptidoglycan synthesis. These essential divisome proteins are reported to be directly involved in (the regulation of) sPG biosynthesis, while other divisome components are allocated additional roles as recruiters of regulators. These include FtsK, FtsA, and FtsEX, which appear to be involved in different stages of divisome assembly and maturation [6,17,18]. This shows that both the recruitment and activity of peptidoglycan synthesis proteins are tightly regulated by numerous factors to form a septum at the appropriate time and location.

In the last decade, great progress has been made in elucidating septal peptidoglycan synthesis. As recent as three years ago, it was reported that the integral membrane protein FtsW possesses glycosyltransferase activity, implying that it is a division-specific PG polymerase [19]. FtsW is only functional when in complex with the bitopic membrane transpeptidase FtsI. FtsW is part of the SEDS (shape, elongation, division and sporulation) family, that also contains the elongation-specific glycosyltransferase RodA [20]. FtsI is part of the unrelated family of class B PBPs (penicillin-binding proteins) that also contain PBP2, which interacts with RodA [21]. These class B PBPs are monofunctional transpeptidases, whereas the related class A PBPs possess both glycosyltransferase and transpeptidase activity. It was recently reported that class A PBP1b is in complex with FtsWI, forming the peptidoglycan synthesis machinery during division [15,22].

While it has already been established that the synthesis machinery is recruited by FtsBLQ, it was recently reported that the recruitment is mostly dependent on a cytoplasmic interaction between FtsL and FtsW [15,23,24]. The direct regulation by FtsBLQ is clouded in mystery compared to its recruiting capacities. In vitro enzymatic assays regarding the regulatory role of FtsBLQ report that the different subunits affect sPG synthesis in different ways [25,26]. The ambiguous in vitro data describe both suppressing and activating properties of the FtsBLQ subcomplex, suggesting that the subcomplex switches between active and inactive conformations and accordingly affects the synthesis machinery. The activating protein FtsN is proposed to play an important role in the switch from an inactive to active state of the PG synthesis complex. FtsN was identified early on as a trigger for division, but the exact mechanism has only recently been partly uncovered [27]. It has been reported that the small essential domain of FtsN (^E^FtsN) alleviates the FtsBLQ-mediated suppression of PBP1b and accelerates PBP1b GTase activity [28].

Furthermore, in recent years, other processes have been discovered that indirectly affect the synthesis of sPG. This includes the discovery of a PG synthesis track moving independently from FtsZ-ring dynamics. The synthesis track is indicated by multiple studies where the dynamics of FtsN and FtsWI follow the rate of sPG synthesis, which differs from FtsZ treadmilling dynamics [29,30]. Interestingly, the FtsZ-ring appears to be involved in the spatial distribution of the synthesis track, which is proposed to be linked to the activity of the semi-essential FtsEX subcomplex. It was recently reported that FtsEX competes with FtsA for the same region of FtsZ, indicating that FtsEX is guided by FtsZ treadmilling over the division site [5]. FtsEX also activates the amidase activity of AmiA and AmiB through EnvC, which results in denuded glycans that can be bound by the Sporulation-related repeat (SPOR) domain of FtsN [31]. This suggests that the activity of FtsEX leads to the acceleration of sPG synthesis through FtsN accumulation and spatial distribution of the synthesis track over the division track.

Additionally, the recruiting capacities of FtsA and the partial redundancy of FtsN are discussed. Not too long ago, FtsA was thought to be directly involved in sPG synthesis through the activation of FtsW [32]. However, here we argue that superfission activity induced by FtsA mutants appears to be linked to premature FtsN recruitment by FtsA, rather than the direct activation of FtsW [33]. Furthermore, the discovery of another SPOR domain-carrying protein that is involved in division indicates the partial redundancy of FtsN. DedD is the only other SPOR domain-carrying protein, next to FtsN, that unconditionally results in division defects when absent, at least in *E. coli* [34,35]. The potential parallel role of DedD in the activation of the sPG synthesis machinery is discussed in detail.

This review will summarize and discuss recent data on the regulation of septal PG synthesis. Based on the reported mutants, structural models have been made of the various protein complexes involved using AlphaFold 2 [36,37,38]. Based on the collected information, a new model on the organization of the divisome is presented.

## 2. Peptidoglycan Biosynthesis and Remodeling

The biosynthesis of new peptidoglycan is a well-known process and has been explored in detail through the years (Figure 2a). The synthesis of peptidoglycan starts in the cytosol, where fructose-6-phosphate is converted to UDP-*N*-acetylglucosamine (UDP-NAG) and subsequently UDP-*N*-acetylmuramic acid (UDP-NAM), which has been extensively reviewed in the past [39]. These precursors are further processed to membrane-bound Lipid II (C_55_-PP-NAM-NAG) and flipped over to the periplasmic side of the cytosolic membrane [40,41,42]. Lipid II is then incorporated in an already growing peptidoglycan chain by PG glycosyltransferases (GTs), forming a linear carbohydrate polymer [43]. New PG chains are then cross-linked through the peptide chains attached to the NAM residues by peptidoglycan transpeptidases (TPs), resulting in the typical mesh-like structure of peptidoglycan [44].

However, to insert new subunits, the covalently closed peptidoglycan layer must be opened by hydrolysis of certain bonds in the peptidoglycan layer (Figure 2b). The PG hydrolases are grouped based on their catalytic activity: (i) PG amidases which cleave off peptide chains attached to NAM residues; (ii) PG glucoasminadases and lytic transglycosylases that hydrolyze and cleave the glycosidic linkage between adjacent NAG and NAM residues, respectively; (iii) PG peptidases which cleave amide bonds in the cross-linked peptide chains [45]. PG hydrolases create space within the PG and thus play an important role during the synthesis and remodeling processes, such as in cell division [46]. PG hydrolases and lytic transglycosylases are reported to drive the separation of the daughter cells, and amidases in particular have been indicated to be involved in the spatial distribution of the sPG synthesis complex [8,47,48,49]. Both PG synthesis and hydrolysis are tightly controlled and coordinated during the whole cell cycle, as severe deregulation results in cell lysis [14].

The presence of two distinct complexes allows cells to separately control peptidoglycan synthesis during elongation and division, resulting in the well-known rod-shape. RodA, PBP2 and PBP1a are the PG-synthesizing proteins associated with the elongasome, while the earlier mentioned complex of FtsW, FtsI, and PBP1b synthesizes PG during division. The conserved structure of both the essential SEDS (RodA/FtsW) and their associated class B PBP (PBP2/FtsI) indicates that the activating mechanism of RodA-PBP2 is similar to that of FtsW-FtsI [20,21]. Recently, the crystal structure of the RodA-PBP2 complex has been produced [50]. The structure reports an interaction between the pedestal domain of PBP2 and a large periplasmic loop of RodA called ECL4 that leads to the activation of RodA GTase activity. Based on mostly genetic studies, a similar model is produced for the FtsW-FtsI complex in the next section of this review.

## 3. The Septal Peptidoglycan Synthesis Machinery

As aforementioned, the sPG synthesis machinery is the part of the divisome that synthesizes new peptidoglycan at the division site in an inward manner during cell division, creating a septum between the two newly formed daughter cells. Septal peptidoglycan synthesis involves the same processes as during general PG biosynthesis, needing both glycosyltransferase (GT) and transpeptidase (TP) activity. The SEDS family member FtsW was designated the role of a division-specific flippase, until a recent study provided in vivo and in vitro evidence of GT activity by FtsW [19]. FtsW forms a stable ternary complex with the essential transpeptidase FtsI (also called PBP3), and the non-essential bifunctional PBP1b, at least in *E*. *coli* [15,51].

### 3.1. FtsWI Holoenzyme Is Exclusively Involved in Cell Shape Remodelling, While PBP1b May Only Play a Minor Role during Division

The complex formed by the FtsWI and PBP1b establishes redundancy in the septal peptidoglycan machinery, as both PBP1b and FtsWI possess GT- and TPase activity [52]. However, while PBP1b is not essential due to (partial) redundancy with PBP1a, FtsW and FtsI are vital for division, indicating additional roles for FtsW and/or FtsI [53]. FtsW recruits FtsI, and both are required for the midcell recruitment of downstream divisome components such as the putative lipid II flippase MurJ and sPG synthesis activator FtsN [54,55,56]. Furthermore, the depletion of either FtsW or FtsI causes multiple mislocalized FtsZ rings and arcs in the ovoid-shaped *S*. *aureus*, as was observed with SIM microscopy [57]. While Mercer and Weiss (2002) [55] showed no effect of FtsW depletion on FtsZ ring abundance in *E*. *coli* with epifluorescence microscopy, a super-resolution method such as 3D-SIM microscopy might uncover similar FtsZ ring mislocalizations.

Moreover, FtsW inhibits PBP1b GT- and TPase activity in the absence of FtsI, at least in vitro [15]. This has been attributed to lipid II retention by FtsW, as the GTase activity of FtsW is dependent on the interaction with FtsI [19]. The activity state of FtsW could act as a rate-limiting factor in vivo, either through lipid II retention, or through a conformational change affecting PBP1b. Allosteric interactions between FtsW and PBP1b may be possible, as co-expression experiments indicated a synthesizing subcomplex assembled around FtsW [15]. Furthermore, a recent publication reported that class A PBPs such as PBP1b are mostly involved in cell wall defect repairs and not PG synthesis, which suggests that the FtsW-FtsI holoenzyme is exclusively involved in cell shape remodeling, while PBP1b may only play a minor role during division [58]. This may seem surprising, as PBP1b is the only sPG-synthesizing protein that is known to be directly activated by the essential activator FtsN [26,28]. PBP1b GTase activity can be activated in vitro by both LpoB and FtsN, but only LpoB can activate PBP1b TPase activity [28,59]. This suggests that the LpoB-mediated activity of PBP1b may increase the mechanical integrity of PG through PG crosslinking, while the FtsN-mediated activity may only contribute to the amount of sPG. Major depletion of PBP1b only results in reduced mechanical integrity, but complete PBP1b deletion halves the total PG amount, suggesting that only a small amount of PBP1b is involved in novel PG synthesis [58,60]. The bulk of PBP1b proteins may repair cell wall defects, which may readily occur at the septum due to the high concentration of PG hydrolases and lytic transglycosylases [61,62,63]. PBP1b may thus have a primary role as a repair enzyme and a minor role as a sPG-synthesizing protein. Altogether, this suggests that the redundancy seen at first sight is negated by the additional roles of FtsW-FtsI and PBP1b. The focus in this review lies on FtsWI, due to their necessity during cell division, though PBP1b will also be discussed as part of the synthesis machinery.

FtsI and FtsW are essential proteins that are recruited to the division site and seem to have a stabilizing effect on the divisome [54,64,65]. The septal recruitment of FtsI depends on FtsW and the deactivation of either results in severe elongation in the case of rod-shaped bacteria, or enlargement and septal abnormalities in the case of spherical bacteria [57]. FtsW is in an intimate relationship with FtsI, and the specific interaction between these two is important for the activity of the sPG synthesis machinery. Here, the relationship between FtsW and FtsI is further elucidated.

### 3.2. The ECL4 Loop of FtsW Regulates GTase Activity through Active Site Modulation

While the crystal structures of both FtsI and PBP1b have been determined, no such analysis has yet been performed for the FtsW structure [44,66]. Luckily, advances in the prediction of protein structures have made it possible to simulate the structure of FtsW and its interaction with FtsI with high accuracy [36,37,38]. As the FtsWI structure is presently not available, the FtsWI structure was simulated using AlphaFold as part of this review (Figure 3). It has been suggested in the past that FtsW GTase activate may be the main target for sPG synthesis regulation, as FtsI TPase activity is important for the structural integrity of the sPG mesh, but not for the synthesis of new PG strands [25]. While this has yet to be proven, there are indeed genetic analyses indicating that this might be the case. The locations of these activating or inhibiting mutations in the predicted FtsW structure show an interesting pattern that could suggest a possible activating mechanism mimicking that of RodA (Figure 4). The most recent report describing key active site residues has been a breakthrough, as this provides a way to deduce a certain regulation mechanism [67]. Similarly, as with RodA, the FtsW active site is predicted to form a groove with the putative entrance between TM2 and TM3, leading to a cavity in the center (Figure 4; roughly the purple circle) [68]. The putative catalytic residue D297 resides on a long periplasmic loop between TM7 and TM8, while the bulk of the active site is in other regions (Figure 4). This periplasmic loop is also called ECL4, as it is the fourth extracellular loop.

Interestingly, ECL4 contains multiple residues involved in FtsW activation, which are close to a periplasmic loop between TM9 and TM10 essential for FtsI midcell recruitment [65]. The effect of this region on FtsW is ambiguous, with both activating and inhibiting properties. The region can be roughly divided in a potential inhibiting region around residue E289 (Figure 4, red dotted ellipse), and an activating region around residue M269 (Figure 4, green dotted ellipse). E289 might regulate FtsW activity through a predicted interaction with residue R73, which is located between TM1 and TM2. Replacing the negatively charged glutamic acid (E) residue for a glycine (G) or a non-polar isoleucine (I) leads to a constitutively active form of FtsW that (i) suppress FtsI mutants defective in sPG/FtsW activation, (ii) partly bypasses FtsN, and (iii) rescues dominant negative FtsL mutants [69]. This indicates that the negative charge on residue 289 is important for the suppression of FtsW activity in the absence of FtsI (and perhaps other activators). E289 is located on the same subloop as D297, which is predicted to be an unstructured loop between the second and third α-helix of ECL4. The E289-R73 interaction may lead to the withdrawal of catalytic residue D297 from the active central cavity, leaving FtsW in an inactive conformation (Figure 4). While still a putative interaction, it can easily be validated via a mutagenic analysis where the charge of R73 is removed or swapped, which should result in an active FtsW mutant.

Contrarily, the region around M269 appears to lead to an active conformation. The hydrophobicity of this region seems necessary for its function, as more hydrophobic substitutions (isoleucine; I) of M269 and A270 lead to the rescuing of inactive FtsL mutants and the partial bypassing of FtsN, while more hydrophilic substitutions (K, E or T) lead to an inactive state of FtsWI that can still be recruited to the septum [69]. Less hydrophilic substitutions such as A270T can still be rescued by at least active FtsI or FtsB mutants, while a substitution for a charged residue such as M269K can only be rescued by an additional E289G mutation. M269 and A270 are located on the second α-helix of ECL4 (Figure 4, in green), which is predicted to be spatially close to the third helix just downstream of the catalytic residue D297. The second helix is predicted to be partly above the third helix, which may lead to the deeper burying of the catalytic residue in the central cavity. Indeed, this would support an activating role for the hydrophobic residues M269 and A270 by the stabilization of active residue D297 in the active site.

### 3.3. The Pedestal Domain of FtsI Is Involved in sPG Synthesis through the Activation of FtsW GTase Activity

FtsI consists of a small cytoplasmic domain, a transmembrane domain, a periplasmic non-catalytic pedestal domain, and a transpeptidase domain (Figure 3) [66]. The transmembrane domain interacts with FtsW and is thus essential for midcell localization. The transpeptidase domain is located distally from the membrane in the periplasm and contains the active site for transpeptidase activity [70]. The exact role of the non-catalytic pedestal domain has been long debated, and appears to be involved in the regulation of FtsW activity. Especially the region close to the membrane comprised of an α-helix (α2) and a three-stranded β-sheet (β1) seems to be involved (Figure 5 in cyan). Amino acid substitutions in either the α-helix (G57D, S61F and L62P) or the β-sheet (R210C) result in filamentous growth and the loss of FtsN recruitment, while the midcell recruitment and β-lactam-binding capacities of these mutants is not affected, suggesting that the transpeptidase domain can still function [56].

Thus, it was long assumed that the pedestal domain was primarily involved in FtsN recruitment. However, a recent study found the mutation K211I next to the division-defective R210C mutation could bypass FtsN, indicating that this region is involved in FtsWI activation [69]. G57D, S61F, and R210C mutants can be rescued by adding the K211I mutation, indicating that the positively charged K211 inhibits sPG synthesis, possibly through suppression of FtsW activity. Based on protein–protein interaction simulations, K211 on the second strand of the three-stranded β1 domain most likely interacts with FtsI residue Q65 on its first strand (Figure 5, residues in red). The K211–Q65 interaction may lead to the stabilizing of β1, while the loss of stability between the strands may lead to the activation of sPG synthesis/FtsW. Overall, these mutational studies show that the proximal part of the pedestal domain is directly involved in FtsW activation. This would be in line with the role of the proximal part of the pedestal domain of PBP2 that activates RodA. However, the FtsI K211I mutant indicates that the interaction between the proximal pedestal domain and FtsW may only activate FtsW to baseline levels, which is then accelerated (or suppressed) by other effectors.

As with K211I, the more distant R167, which may interact with E193, can partly bypass FtsN when replaced with a serine residue (S) (Figure 5, residues in red). However, substitution by a glutamine residue (N) leads to diminished penicillin binding and division defects [29,71]. It may be that pedestal domain instability caused by R167S substitution leads to a constitutively active form of FtsWI, while R167N substitution, which was combined with a R166Q substitution in the study, severely destabilizes the transpeptidase domain and thus diminishes FtsI activity. Interestingly, both the structural prediction as well as the crystal structure of FtsI show an exposed hydrophobic patch (V84, V86 and V88) close to R167 [66] (Figure 5). Residue V86, when substituted for a glutamic acid (E), cannot activate sPG synthesis but still localizes to midcell and binds to penicillin [56]. The deletion of a loop (150–161) close to a similar region in PBP2 largely maintained RodA GTase activity, indicating that the region around R167 may not directly be involved in activating FtsW [50]. Based on the mutational data available for FtsI mentioned above, we argue that the distal region of the pedestal is involved in the additional stimulation of FtsW when in contact with FtsL through a hydrophobic interface. This is discussed further in Section 4.3.4 of this review.

Overall, the FtsI pedestal domain appears to be involved in the activation of sPG synthesis, likely by regulating the GTase activity of FtsW. The fact that in vitro FtsW GTase activity can be activated by a catalytic inactive FtsI variant indicates that the TPase domain is not implicated in FtsW activation [19]. In vitro evidence of the FtsW-activating properties of FtsI residues G57, S61, L62, and R210 is lacking, but can be validated with assays developed in previous years [19,25,26]. If validated, this could mean that FtsI activates FtsW GTase activity to a baseline level, which is then amplified or suppressed by other regulating factors. Though there is some in vitro evidence supporting an amplifying role for regulators, evidence for both mechanisms is indirect, so much remains to be discovered to reach a convincing conclusion [25]. For now, the evidence points to an activating role for the proximal part of the pedestal domain, while the distal part may be involved in the acceleration of FtsW activity.

### 3.4. sPG Activity May Be Regulated by the Modulation of the Interaction between FtsW and FtsI

A figurative tug-of-war for the putative catalytic D297 residue between the region around M269 and the region around E289 could be a possible mode of regulation for FtsW (Figure 6). An interaction between the M269 region and the FtsI proximal pedestal domain may stabilize the catalytic residue D297 in the central cavity. Contrarily, the interaction between FtsW residues E289 and R73 may destabilize the catalytic residue, diminishing FtsW activity. Certain interactions within the FtsI pedestal domain such as K211–Q65 and R167–E193 appear to have a similar effect, suppressing FtsW activity. Indeed, this indicates that the interaction between the FtsI pedestal domain and FtsW ECL4 determines the state of sPG synthesis.

Regulators may modulate the stability of this interaction, as it is determinative for FtsW GTase activity [19,25]. This is in line with the proposed activation mechanism of RodA by PBP2 [50]. Although the predicted structure of the FtsI-FtsW complex is still putative, it is like the crystal structure of PBP2-RodA. Both the TM domains of PBP2 and FtsI interact tightly with RodA/FtsW TM domains 8 and 9, and both the PBP2/FtsW pedestal domains interact with ECL4 of RodA/FtsW. The only major difference is that RodA-PBP2 complex formation leads to the outward deflection of TM7, creating a membrane-accessible cavity. This cavity was observed in both the structural prediction of FtsW separately and in complex with FtsI. Additionally, Sjodt et al. reported a direct interaction between the PBP2 distal pedestal domain and RodA ECL4 close to TM7, which was not predicted. These differences may be due to the crystal packing effect, as the RodA-PBP2 complex adopts numerous conformations in electron microscopy data [50,72]. In the next section, we present a model where regulators allosterically interact with and stabilize FtsWI, and show that especially an interaction between the distal pedestal domain of FtsI and FtsL is important for accelerated sPG synthesis.

## 4. Septal Peptidoglycan Synthesis Regulation

The sPG synthesis machinery is recruited and regulated by certain factors localized at the midcell. These factors include the conserved subcomplex of the divisome consisting of FtsB, FtsL and FtsQ, and the septation-activating protein FtsN. The FtsBLQ subcomplex and FtsN are involved in the recruitment and regulation of the septal peptidoglycan synthesis in a diverse subset of bacteria including the rod-shaped *Bacillus*, *Escherichia* and *Pseudomonas* genera, and the ovoid-shaped *Staphylococcus* and *Enterococcus* bacteria [16,25,73]. The essentiality of the FtsBLQ subcomplex and FtsN, known as the depletion or inactivation of FtsB, FtsL, FtsQ and FtsN, results in extensive cell filamentation in *E. coli* [17,74,75]. The essentiality of at least FtsL and FtsN is conditional, as (activating) mutations in other divisome components can complement for their depletion or inactivation [23,69]. FtsBLQ and FtsN have no enzymatic activities and are thus assigned the role of regulators through allosteric interactions.

Two modes of regulation are currently proposed. The first mechanism is entirely based on inhibition through the interaction of FtsBLQ with the sPG synthesis machinery, which is lifted by the direct interaction between sPG synthesis activator FtsN and the machinery (the inhibition model). However, (muta)genetic analyses propose a model based on active and inactive conformations of the FtsBLQ complex triggered by direct interaction with certain domains of FtsN (the ON/OFF model). Here, we will discuss the individual components of sPG regulation and propose a model that best matches the collected data. This includes the direct regulatory interactions of FtsN and FtsBLQ, the supposed activation of sPG synthesis by FtsA, and the accelerating role of FtsEX.

### 4.1. FtsA: The FtsN Recruiter

FtsA is one of the more surprising proteins that is suggested to be directly involved in the activation of sPG synthesis. FtsA is a partly membrane-embedded cytosolic protein that acts as a membrane anchor for FtsZ [76]. It is structurally related to eukaryotic actin forming comparable protofilaments, and it has been reported that oligomerization of FtsA plays a role in the regulation of division [77,78]. A model has been proposed in which FtsA forms minirings comprised of 12 subunits that align FtsZ filaments and prevent FtsZ filament bundling, thus delaying division till the activation signal arrives [79]. Although the evidence for FtsA minirings is all based on in vitro data, it is an interesting idea that links FtsA to the delay of divisome assembly. FtsA has indeed been linked to the recruitment of downstream factors in a FtsEX-dependent manner. This likely happens through the disruption of the interaction between a short conserved C-terminal region of FtsZ and conserved residues in the FtsA 2B subdomain by FtsEX (Figure 7 in blue) [5,80]. This disruption may lead to FtsZ filament bundling and the subsequent recruitment of late divisome proteins.

The involvement of mostly cytoplasmic FtsA in sPG synthesis regulation is a remarkable feature compared to the other bitopic, mostly periplasmic regulators. No in vitro evidence of a direct link between FtsA and sPG synthesis exists. However, in vivo data suggest an activation signal relayed through FtsA [16,32]. Superfission (SF) mutants of FtsA and FtsB or FtsL can synergistically rescue complete FtsN depletion [16]. Additionally, the overexpression of SF mutant FtsA^R286W^, in contrast to FtsN overexpression, can partly complement for FtsQ depletion [32]. Furthermore, FtsA SF mutants can partly bypass FtsL dominant-negative (DN) mutants. This all indicates that FtsA is indeed involved in the activation of FtsWI.

Unfortunately, it is not yet clear whether FtsA directly activates sPG synthesis. In vitro data examining the effect of FtsA on the enzymatic activity of FtsWI is presently not available. Overall, the available data indicate that FtsA SF mutants rescue cell division through a back-recruiting mechanism involving FtsN. A back-recruiting interaction between FtsN and FtsA is essential in situations when late protein recruitment is disrupted, such as in the absence of FtsEX or FtsK [18,82]. This also likely occurs in the case when FtsQ is depleted, where the FtsA^R286W^ SF mutant acts as bait for FtsN. Similarly, FtsA^R286W^ can rescue FtsB and FtsL mutants that disrupt FtsWI recruitment, but not the complete absence of FtsB or FtsL. This suggests that FtsA may be involved in the midcell recruitment of FtsW, as indicated by the FtsA–FtsW interaction reported by a BACTH assay [32].

Back-recruitment by FtsA appears to be the most likely option for now, as the evidence for FtsW activation by FtsA is weak. Firstly, FtsA SF mutants can only rescue a less defective FtsW mutant still responsive to FtsN overexpression, but not a non-functional FtsW mutant [69]. This suggests that the FtsA-mediated activation of FtsWI is dependent on FtsN. Secondly, FtsA SF mutants can recruit FtsWI independently of FtsQ, FtsL^cyto^ or the FtsB periplasmic domain [32]. However, the bypassing of FtsBLQ in the recruitment of FtsWI occurs in the presence of FtsN. This could mean that the FtsA SF mutants recruit FtsWI indirectly by prematurely recruiting FtsN. Lastly, the abovementioned papers mostly studied the rescuing of cell growth on spot assays and not the actual recruitment or activation of FtsWI. The role of the activating interaction between the essential ^E^FtsN domain and PBP1b (see Section 4.2) is not considered in these studies, together with additional regulating factors. This may explain why FtsA SF mutants can partly bypass FtsL DN mutants, as their increased recruitment of FtsN may activate sPG synthesis despite suppressing FtsL mutants.

The FtsN–FtsA interaction is enhanced when FtsA is not able to self-interact [18,83]. The known FtsA SF mutants are clustered in the oligomerization interfaces (Figure 7, red residues). This supports the notion that FtsA SF mutants lead to an increased and premature recruitment of FtsN. Mutations in the FtsN-interacting FtsA 1C domain that suppress non-functional ^E^FtsN mutants are still dependent on the mutated FtsN protein or additional FtsB or FtsL SF mutants. This suggests that FtsA SF mutants recruit FtsN, which then activates FtsWI-PBP1b, or leads to back-recruitment when FtsWI recruitment is disrupted. This has indeed been shown in a recently published study, where FtsA^R286W^ led to the earlier recruitment of FtsN, but had no effect on the onset of constriction [84]. Altogether, this indicates that FtsA is primarily involved in the recruitment of FtsN, and not directly in the activation of the sPG synthesis machinery.

### 4.2. The Activator FtsN

FtsN is regarded as the trigger for constriction of the cell during division. The protein consists of a small cytoplasmic domain, a transmembrane domain, and a large periplasmic domain containing a PG SPOR domain (Figure 8). FtsN interacts with FtsA in the cytosol and with FtsQ in the periplasm, which indicates that it is partly recruited by these interactions [33,85]. Indeed, FtsN recruitment requires at least both FtsA and FtsQ, as was shown by the ‘premature targeting’ of different late divisome proteins [24]. Furthermore, FtsA and FtsQ both interact with the late-divisome members FtsW or FtsI, which may change FtsA and FtsQ to an FtsN-recruiting conformation [26,32]. This suggests that these interactions signal that the divisome is completely assembled at both sides of the cytosolic membrane, and thus septation can commence. Interestingly, only a small part of the periplasmic domain (^E^FtsN domain; Leu^75^-Gln^93^) appears to be essential, as deletion of the other domains only delays septation, but does not prevent it [86,87]. This is likely linked to the functionality of the SPOR and cytoplasmic domain, which are involved in the septal accumulation of FtsN and the initial recruitment by FtsA, respectively [82,88]. ^E^FtsN interacts in vitro with the GTase and UB2H domain of PBP1b to activate its GTase activity, but does not interact with FtsW, FtsI or FtsBLQ, as was shown in a fluorescence anisotropy assay [28].

#### 4.2.1. Partial Redundancy of FtsN by DedD

It is puzzling how a domain of only 18 residues, which only activates the GTase activity of PBP1b, can be essential for cell division in *E*. *coli*. It had been shown in vitro that full-length FtsN activated and alleviated the FtsBLQ-mediated inhibition of PBP1b GTase activity, although a similar study with *Pseudomonas aeruginosa* FtsN did not show such an effect on FtsW activity [25,26]. As an activator of the sPG synthesis machinery, one would expect that FtsN would also activate the FtsWI pair; however, such a mechanism has not yet been discovered. It has been shown in vitro that the combination of FtsWI-PBP1b and FtsN increased GTase activity, suggesting some kind of allosteric effect [26].

However, the discovery of FtsN homologue DedD in *E*. *coli* might make the activation of FtsWI by FtsN less essential [27]. DedD and FtsN are the only SPOR-carrying proteins in *E. coli* known to unconditionally result in division defects (cell elongation and chaining) when depleted, and are thus directly involved in cell division [35]. The absence of DamX, another SPOR-carrying protein in *E. coli*, is reported to exacerbate the cell chaining phenotype of DedD mutants, but no apparent division defects have been found in single Δ*damX* mutants [27,34,35]. RlpA, the fourth and last SPOR-carrying protein in *E. coli*, acts as a lytic transglycosylase in *Pseudomonas aeruginosa* and *Vibrio cholerae* [48,49]. Δ*rlpA* mutants of *P. aeruginosa* and *V. cholerae* result in a cell-chaining phenotype under specific conditions (i.e., low osmotic strength), while Δ*rlpA* mutants in *E. coli* do not show such division defects [27,35,89]. Thus, DedD and FtsN appear to be the only two SPOR-carrying proteins in *E. coli* directly involved in the activation of sPG synthesis. Moreover, it was long thought that the superfission (SF) mutants of FtsA, FtsB or FtsL could directly activate sPG synthesis without the involvement of FtsN (or a homologue); however, the discovery of DedD calls this into question. While DedD and FtsN cannot complement for the inactivation or deletion of the other, it might be that SF mutants are dependent on DedD to bypass FtsN, and vice versa [34]. Furthermore, the absence of DedD is poorly tolerated in PBP1b-lacking cells, leading to severe cell chaining and septal lysis. This suggests that DedD is at least partially involved in FtsWI activation, as ^E^FtsN, which directly activates PBP1b in vitro, cannot completely activate sPG synthesis in the absence of DedD and PBP1b. Thus, parallel but not identical roles for FtsN and DedD in the activation of the sPG synthesis machinery seem likely.

#### 4.2.2. FtsN Translocates the Synthesis Complex to a Synthesis Track at Midcell

Interestingly, it appears that FtsN has an additional role as a translocator of the sPG synthesis machinery to active synthesis sites (Figure 9). The dynamics during the division of FtsZ and FtsA differ from those of FtsW, FtsI and FtsN. The speed of FtsZ filaments and FtsA proteins has been reported to be close to 30 nm/s, while the average speed of FtsWI is closer to 20 nm/s, and FtsN moves even slower at roughly 9 nm/s [29,30,90,91]. FtsWI appears to have slow and fast-moving populations that respectively correlate with FtsN and FtsZ treadmilling dynamics. The fast-moving population is coupled to FtsZ treadmilling dynamics, while FtsN promotes the slow-moving population [29]. The dynamics of the slow-moving population and FtsN are affected by the rate of sPG synthesis, as the inhibition of sPG synthesis diminishes FtsN movement and the slow-moving FtsW population [29,30]. Overall, this suggests a fast-moving assembly track around the septum controlled by FtsZ treadmilling, and a slow-moving synthesis track promoted by FtsN.

The existence of physically separated subcomplexes assembled around FtsZ and FtsN has been hinted at before, as the radial and circumferential separation of FtsZ and FtsN has been observed [92]. One may hypothesize that when the complete PG synthesis complex is assembled on the FtsZ ring, the complex recruits FtsN and is transported to the synthesis site, which would likely be where the PG layer would have been made accessible by PG hydrolases that produce denuded glycan strands, which are tightly bound by the FtsN SPOR domain [47,88]. The domains of FtsN do show different dynamics when separately expressed. The cytoplasmic domain follows FtsZ-treadmilling dynamics due to an interaction with FtsA, while a functional ^E^FtsN domain moves at speeds comparable to the slow-moving population. The SPOR domain appears to prevent the fast movement of FtsN, and it has been suggested that the SPOR domain contributes to the stationary fraction of FtsN [30]. However, this is not indicated by the available data.

The SPOR domain also plays a role in the septal recruitment of FtsN, likely in enhancing the septal accumulation of FtsN [27]. As mentioned before, the SPOR domain binds to denuded glycans. These denuded glycans are formed by PG amidases AmiA and AmiB, which are activated by the divisome member FtsEX, and also by AmiC, which is activated by the Tol-PAL system through NlpD [5,93,94]. The endopeptidase PBP4 has recently been discovered to localize also in a FtsEX-dependent manner to the division site, and the depletion of PBP4 significantly delays the recruitment of FtsN [61]. Interestingly, it has been indicated that FtsEX competes with FtsA for FtsZ to recruit downstream proteins, and that its ATPase activity, which drives AmiA/B activity through EnvC, is essential for cell separation when NlpD is inactive [5,31].

This suggests a model where a small portion of FtsN is recruited to the midcell by a complete divisome through FtsA and FtsQ, which then activates FtsEX ATPase activity (Figure 9). Endopeptidase activity and amidase activity mediated by FtsEX may result in the accumulation of more FtsN and accelerated division by a positive feedback loop. It could be that the accumulation of more FtsN leads to the translocation of the sPG remodeling complex to the slow-moving synthesis track away from the assembly track. The route of the synthesis track may be carved out by PG hydrolases activated by FtsEX, which is most likely guided around the division site by FtsZ filament treadmilling. This would be in line with the notion that FtsZ treadmilling only affects the spatial distribution, and not the rate of sPG synthesis [95]. A recently published study indeed showed two temporally distinct stages of FtsN recruitment, where the first recruitment stage started septal peptidoglycan synthesis and the second stage accelerated PG synthesis [84]. While the initial recruitment of FtsN was attributed to FtsA-mediated recruitment, the cause of the second stage is not yet established. An interesting theory for the latter may be the abovementioned mechanism of denuded glycan-mediated FtsN accumulation, which would in turn lead to the acceleration of sPG synthesis. However, the glycan-mediated accumulation seems to only play an accelerating role, as the ^E^FtsN domain is sufficient in promoting the slow-moving track through the activation of sPG synthesis [30].

Overall, FtsN plays an essential role as the activator of septal peptidoglycan synthesis, which can only be bypassed by activating mutations in other divisome components. While other potential activators such as DedD are also present, the majority of sPG synthesis appears to be controlled by FtsN. Protein–protein interaction predictions were conducted with FtsWI and FtsN in this review; however, these did not result in clear interaction interfaces. Furthermore, data regarding FtsN complex formations are ambiguous, as FtsN binding partners differ depending on the used technique [26,85,96,97]. Thus far, evidence shows that FtsN directly activates PBP1b through the small ^E^FtsN domain and indirectly activates FtsWI by inducing conformational changes in FtsBLQ [23,28]. In the next section, the latter will be discussed in the context of regulation by FtsBLQ. The next section also discusses the seemingly non-essentiality of a large part of FtsN.

### 4.3. The Regulatory Subcomplex FtsBLQ

The FtsBLQ subcomplex is highly conserved and thought to be the direct regulator of sPG synthesis during division. The subcomplex consists of the three bitopic inner membrane proteins FtsB, FtsL and FtsQ, and plays an essential role during divisome assembly and sPG synthesis regulation (Figure 10). It has been reported that the complex is in a 2:2:2 stoichiometry with FtsB and FtsL forming a tetramer through interactions between their transmembrane domains, and a periplasmic interaction between FtsB and FtsQ, completing the subcomplex [98,99,100]. The role of the subcomplex during sPG synthesis regulation is still ambiguous, as both inhibiting and activating properties have been reported for (subunits of) the FtsBLQ subcomplex. In this section, a model based on in vitro data and extensive genetic analyses is proposed, where the FtsBLQ subcomplex stabilizes the sPG synthesis machinery in (in)active conformation. In this model, instability especially in the FtBL coiled coil is determinative for the state of septal peptidoglycan synthesis.

#### 4.3.1. A Cytoplasmic Interaction between FtsL and FtsW Recruits the Synthesis Machinery

The recruitment of the sPG synthesis machinery to midcell is directly dependent on FtsBLQ, though the role of the individual members of the FtsBLQ subcomplex differ in the recruitment. This is apparent when FtsB, FtsL or FtsQ are depleted, which results in extensive filamentation, indicating a defective cell division machinery [55,101,102]. The conserved polypeptide-transport-associated (POTRA) domain of FtsQ is the first major link between the synthesis complex and the rest of the divisome (Figure 11) [103]. The periplasmic POTRA domain, consisting of two α-helixes and a three-stranded β-sheet, is important for the midcell localization of FtsQ through a putative interaction with FtsK [104]. A two-hybrid screening reported a FtsI/FtsN interaction site containing amino acid residues 49–57 for FtsI, S77 and E125 for FtsN, and residues 105–136 for FtsW located in the POTRA domain (Figure 11), although only the interaction between FtsQ^1^^–^^7^ and FtsI has been confirmed by co-immunoprecipitation (co-IP) [85,105]. The C-terminal domain of FtsQ has been reported to be important for the recruitment of downstream proteins, especially FtsBL. The periplasmic interaction between FtsQ and FtsB has recently been described in detail [100]. It has been reported that FtsL does not directly interact with FtsQ in the periplasm, but indirectly through FtsB [98]. Indeed, this is also observed in the predicted structure of FtsBLQ, where no direct interaction between FtsL and FtsQ is found (Figure 10 and Figure 15). This indicates that FtsQ is the first recruited protein, which then recruits FtsBL through an interaction with FtsB.

The main FtsWI recruiting protein is FtsL (Figure 12). Deletion of the cytoplasmic domain of FtsL results in the loss of FtsWI midcell localization [23]. FtsI forms a permanent complex with FtsW and the strong recruiting interaction made between FtsW and FtsL is apparently sufficient for their localization [51,55]. The midcell enrichment of PBP1b may also be linked to this interaction, as it is part of a trimeric complex assembled around FtsW [15]. Interestingly, a periplasmic interaction between FtsI and FtsL has been reported [23]. This interaction can be strengthened by activating mutations in FtsL, which indicates that this interaction is mostly involved in the activation of FtsWI. Moreover, the TM domain of FtsI can localize to midcell independently of its periplasmic domain, showing that the main recruitment is through the FtsW–FtsL interaction [106]. Indeed, this all indicates that the main recruitment manner of FtsWI is through a cytoplasmic FtsW–FtsL interaction, and that an additional periplasmic FtsI–FtsL leads to the activation of FtsWI.

#### 4.3.2. FtsB and FtsL Both Contain a Control Constriction Domain (CCD), While FtsL Has an Additional Activation of the FtsWI Domain (AWI)

Both FtsB and FtsL have certain domains that directly regulate sPG synthesis [23,107]. The control constriction domains (CCD) of FtsB and FtsL are both involved in the suppression of sPG synthesis, while the activation of the FtsWI domain (AWI) accelerates sPG synthesis. Mutating residues in these domains does not affect the recruitment of FtsWI, but leads to a super fission (SF) or dominant negative (DN) phenotype, respectively. Mutations in the CCD domains are A55T, E56A, and D59H for FtsB, and E88K, N89S, G92D, D93G, and H94Y for FtsL, which all bypass the essential domain of FtsN (^E^FtsN) (Figure 13. However, almost all require the N-terminal part of FtsN or another SF mutant of either FtsL, FtsB or FtsA [107]. The only SF mutant that can unconditionally complement for a complete loss of FtsN is FtsB^E56A^. Substitutions of the negatively charged E56 with alanine, valine (both hydrophobic), lysine (positively charged) or glycine (no charge) all bypass FtsN, indicating that the negative charge is essential for stalling division till FtsN arrives [16]. How the FtsB CCD domain and especially residue E56 regulate sPG synthesis is unknown, as predictions of protein–protein interactions do not show an interaction between the sidechain of E56 and residues of FtsL, FtsQ, or FtsI (Figure 10 and Figure 13) [36,38]. The CCD domain of FtsL is likely involved in the suppression of PBP1b GTase activity, which is elaborated further on in the review. Interestingly, CCD domain mutants are conditionally lethal and result in growth defects such as membrane-blebbing and cell lysis. The conditional lethality of FtsB and FtsL mutants only occurs when ^E^FtsN is present, which is not the case for FtsA SF mutants. As discussed earlier (see FtsA section), FtsA SF mutants are most likely involved in the recruitment of FtsN, but not directly in the activation of sPG synthesis. Altogether, this suggests that the FtsB and FtsL CCD mutants lead to the premature initiation of septation in certain conditions, which is exacerbated when at least ^E^FtsN is present. Indeed, this shows that the CCD domains of FtsB and FtsL suppress sPG synthesis until FtsN arrives.

The AWI domain of FtsL leads to the activation of sPG synthesis. It is located on the same helix as the FtsL CCD domain and partly overlaps (Figure 13). The AWI domain is based on DN mutants that suppress cell division but are still recruited to midcell. To date, the following DN FtsL mutants have been discovered close to the CCD domain: R82E, N83K, L84K, L86F, E87K and A90E [23]. Most of these mutants can be suppressed either by the overexpression of FtsN or the SF FtsL^E88K^ mutant, except for L86F and E87K. The latter two can be bypassed by the overexpression of active forms of FtsW (FtsW^M269I^ and FtsW^E289G^), indicating that especially L86 and E87 are directly involved in the transmission of an activating signal from FtsN to FtsWI, as will be discussed in Section 4.3.4. Furthermore, two FtsL DN mutations outside the AWI domain have been reported by Park et al., FtsL^L24K^ and FtsL^R61E^ [23]. These two mutants produce a weaker DN phenotype than the AWI domain mutations, and thus seem not to be directly involved in the activation of FtsWI. However, these residues appear to somewhat stabilize FtsW, which may be beneficial for sPG synthesis (Figure 12c,d). Indeed, both the CCD domains and the AWI domain are involved in the regulation of sPG synthesis. The CCD domains likely target PBP1b to suppress sPG synthesis, while especially the FtsL AWI domain stabilizes FtsWI in a hyperactive conformation.

#### 4.3.3. PBP1b Activity Is Suppressed by the FtsL CCD Domain and Activated by ^E^FtsN

Despite being a non-essential part of the synthesis machinery, the regulation of the class A PBP1b during division has been mostly uncovered in recent years. It was earlier reported that FtsBL in vitro inhibited PBP1b GTase activity [26]. While the FtsB SF mutants E56A/K did not increase PBP1b activity, FtsL SF mutant D93A decreased the repressing effect of FtsBLQ on GTase activity of PBP1b, which points to a direct suppressing interaction between FtsL and PBP1b. The suppression of PBP1b GTase activity is also alleviated by the ^E^FtsN domain, as reported by multiple in vitro studies [26,28]. If the CCD domain of FtsL inhibits PBP1b activity in vivo, and this inhibition is nullified by competition with ^E^FtsN, this would imply that FtsL interacts with PBP1b until FtsN arrives. A PBP1b activating mechanism through competition between the CCD domain of FtsL and ^E^FtsN for PBP1b might be possible, as both domains show some sequence homology (Table 1) [28]. This includes ^E^FtsN residues L89 and E90, which are similar in physiochemical properties to FtsL^CCD^ residues G92 and D93. Charge substitutions of the mentioned FtsL residues lead to an SF phenotype, suggesting less PBP1b suppression by FtsL^CCD^. Substitutions mimicking the ^E^FtsN domain, such as L86F (mimics the bulky FtsN^W83^) or E87K (mimics FtsN^R84^) affect sPG synthesis negatively, perhaps through the increased affinity of FtsL for PBP1b. This provides evidence for an inhibitory FtsL–PBP1b interaction, which can be investigated in a similar manner as the ^E^FtsN–PBP1b interaction [28]. Overall, this suggests that a direct FtsL–PBP1b interaction stabilizes the synthesis complex in an inactive conformation. Interfering with this interaction may change the balance to a more active form, which could occur by allosteric binding of the ^E^FtsN domain.

#### 4.3.4. The AWI Domain Appears to Directly Activate FtsWI through the Allosteric Binding of a Hydrophobic Pocket around FtsI Pedestal Domain Residue V86

The main activating allosteric interaction appears to be the one between the AWI domain of FtsL and the periplasmic domain of FtsI. Certain mutants in the AWI domain (FtsL^L86F^ and FtsL^E87K^) cannot be rescued by FtsN overexpression or active FtsL^G92D/E88K^ mutants, unlike other FtsL DN mutants [23]. The non-rescuable FtsL mutants can be bypassed by the overexpression of active forms of FtsW (FtsW^M269I^ and FtsW^E289G^), indicating that especially L86 and E87 are directly involved in the transmission of an activating signal to FtsWI. Moreover, a mutant lacking the cytoplasmic domain, FtsL^Δ1–30^, combined with an E88K mutation can still rescue and bind FtsWI; however, additional L86F or E87K DN mutations cancel this effect. This may seem counterintuitive, as it was just mentioned that the cytoplasmic interaction is essential for FtsWI midcell recruitment. Notably, recruitment by a periplasmic FtsL occurs only with additional SF mutations, showing that recruitment is mainly controlled by the cytoplasmic domain [23]. Indeed, this indicates that FtsL allosterically interacts in the periplasm with FtsWI.

The periplasmic interaction is most likely between the AWI domain of FtsL and the pedestal domain of FtsI. A stronger interaction has been reported between FtsI and FtsL^Δ1–30/G92D/E88K^ than between FtsW and the same FtsL mutant [23]. Based on protein–protein interaction predictions, the AWI domain interacts with the pedestal domain of FtsI (Figure 12b). A surface-exposed hydrophobic pocket around FtsI^V86^ (V84, V88) may interact with the FtsL AWI domain through residue L86. FtsI^V86E^ and FtsI^G57D^, which are defective in FtsW activation, can be rescued by SF mutant FtsL^G92D/E88K^ [23]. However, both mutants cannot be rescued by active FtsW^M269I^. This supports the FtsW dependance on the FtsI pedestal domain and suggests that the FtsL AWI domain is involved in the stabilization of the FtsWI complex. Overall, an interaction between the FtsL AWI domain and the distal part of the FtsI pedestal domain seems to stabilize the active FtsWI complex, leading to an accelerated sPG synthesis state.

The other two DN mutants outside the AWI domain, L24 and R61, are also involved in the stabilization of FtsW, though they seem less involved in the hyperactivation of FtsWI [23]. FtsL^L24^ is in the cytoplasmic region that recruits FtsW and is predicted to interact with hydrophobic residues in transmembrane domain 1 (TM1), TM2, and TM10 of FtsW (Figure 12d). A FtsW loop between TM1 and TM2 carries R73, which is predicted to interact in an inhibiting manner with FtsW^E289^, and active site residue FtsW^G380^ resides at the periplasmic side of TM10. The cytoplasmic interaction between FtsL^L24^ and the hydrophobic FtsW residues may stabilize FtsW in a more active conformation, as substitution for a more polar residue leads to a weak dominant negative phenotype, which is exacerbated by an additional FtsL^I28K^ substitution. Furthermore, FtsL^R61^, which is in the periplasm close to the inner membrane, is predicted to interact through H-bonds with FtsW^G282^ (Figure 12c). Interestingly, FtsW^G282^ is located on the same periplasmic loop as E289 and the putative catalytic site residue R297. The putative interaction between FtsL^R61^ and FtsW^G282^ may again stabilize FtsW in a more stable (active) conformation, as charge substitution on FtsL^R61^ leads to a mild DN phenotype. Overall, these studies indicate that FtsL stabilizes FtsWI, where especially the AWI domain is involved in the stabilization of the FtsI pedestal domain, thus increasing FtsWI activity.

#### 4.3.5. FtsQ May Inhibit FtsWI Activity by Interacting with the FtsI Pedestal Domain

A major missing aspect is the in vitro measured inhibition of the TPase activity of FtsI by FtsQ, in contrast to FtsBL, which did not affect FtsI activity [26]. While FtsW GTase activity is likely to be the main target of regulation, targeting FtsW through FtsI is a logical possibility, as FtsI is directly involved in FtsW activation [19,23,108]. A FtsBLQ complex with a DN FtsL mutant diminishes in vitro FtsW synthesizing properties to 5–10-fold less than baseline FtsWI levels [25]. As FtsQ has been reported to decrease FtsI activity, this suggests that FtsQ suppresses sPG synthesis through FtsWI [26].

It has been reported that the interaction between FtsQ and FtsI/FtsN mainly depends on residues 50–57 of FtsQ, which are predicted to form an unstructured loop between the inner membrane and the POTRA domain (Figure 14) [85,105]. A second interaction was hinted at for C-terminal FtsQ region 202–227, which forms part of a three-stranded β-sheet and an adjacent α-helix. The interaction of FtsQ^1−57^ with FtsI was confirmed by co-IP, and an earlier study reported a potential interaction between FtsI region 51–70 and FtsQ [109]. FtsI^51−70^ consists of an α-helix and a β-sheet (Figure 14) and is also the region where the bulk of the FtsW-activating residues (G57, S61 and L62) are. Interestingly, the alignment of FtsQ^50−57^ and FtsI^51−70^ suggests an interaction interface consisting of multiple charged and hydrophobic residues (Table 2). The FtsI–FtsQ interaction interface is supported by PBP2 mutations corresponding to FtsI, V54, and K55 that result in an active phenotype [69]. This interface is located between the FtsW-activating portion of the FtsI pedestal domain and a flexible hinge region close to the membrane, suggesting that FtsQ inhibits FtsWI through allosteric binding to this region. Furthermore, the region 105–136 FtsQ is indicated to interact with FtsW region 67–75, which is a periplasmic loop between TM1 and TM2 (Figure 11 and Figure 13) [105]. Interestingly, the FtsW region that is thought to interact with FtsQ contains residue R73, which is proposed to interact with FtsW residue E289 to negatively regulate the GTase activity of FtsW (Figure 4). So, it is possible that FtsQ binds to the FtsWI regulatory interface, thereby suppressing FtsWI activity.

Furthermore, multiple residues have been reported in the periplasmic domain of FtsQ which were essential for the FtsN interaction, but did not affect the FtsI–FtsQ interaction (Figure 11) [85]. These include POTRA domain residues S77 and E125, and the C-terminal residue S242. However, it is difficult to assess whether FtsN directly interacts with FtsQ^50−57^ to lessen inhibition by FtsQ, or if a FtsI–FtsQ interaction at this region allows for FtsN binding. A similar validation by co-IP, as performed with FtsI–FtsQ^1−57^, could be performed to be conclusive about this interaction. FtsQ might change to a FtsN recruiting conformation only after FtsWI binding, as the interaction between FtsQ and FtsW has been reported to be stronger than between FtsQ and FtsN [26]. If FtsN indeed directly interacts with this FtsQ region, the FtsQ-mediated inhibition of FtsWI could be alleviated by a direct interaction between FtsN and FtsQ.

Overall, it appears that FtsQ is involved in the suppression of sPG synthesis by the allosteric inhibition of FtsWI, which may be alleviated by the FtsL AWI domain or FtsN [25].

#### 4.3.6. FtsBLQ (in)Active States Are Modulated by FtsBL (inter)Coiled Coil Instability

The allosteric interactions between FtsBLQ and the synthesis machinery indicate active and inactive conformations of the FtsBLQ subcomplex. Especially the activating properties of the FtsL AWI domain support a conformational change that leads to the interaction between FtsL AWI and the pedestal domain of FtsI. The periplasmic FtsB and FtsL regions mainly form a coiled coil structure (Figure 13) [75]. The modulation of coiled coil stability appears to induce different conformations of FtsBLQ, that either suppress or activate sPG synthesis.

FtsBL is most likely in a tetrameric (2:2) Y-conformation, where each branch of the Y forms a FtsBL coiled coil [110]. It has recently been suggested that contact between the two FtsBL coils is mediated by a mildly hydrophobic alanine patch in FtsB (A37, A38, A41, A44 and A48; Figure 13) [111]. Intercoil disruption seems to lead to smaller cells, suggesting that modulation of coiled coil stability is important for sPG synthesis regulation. The FtsBL coiled coil could be more stable, were it not for a polar cluster in the middle of the structure (Figure 13) [111]. Mutating these polar residues to non-polar (FtsB^Q39L^, FtsB^N50I^, FtsL^R67I^) increases the thermal stability of the coiled coil and leads to elongated cells. Interestingly, the adjacent FtsL^R74E^ mutant produces a SF phenotype, indicating a polar cluster that provides dynamic regulation of the stability of the coiled coil. Concurrently, simulations of FtsBL interactions indicate that the coiled coil regions of FtsB and FtsL only tightly interact in the periplasm around a hydrophobic pocket consisting of residue between the TM and CCD domains (L46 of FtsB, and L77 and W81 of FtsL), while polar clusters in the putative coiled coil destabilize the structure [110]. Thus, the stability of the coiled coil is heavily implicated in sPG synthesis regulation, as both the CCD and AWI domains can be found towards the end of this structure. The CCD domain of FtsB may be involved in stability modulation, as both the modelling of Craven et al. and our protein–protein interaction predictions indicate that it is located at the same helix face as the intercoil region (Figure 13) [111].

The evidence for a stabilizing role of the FtsB CCD domain is mostly indirect. CCD domain residue E56 is the only known residue that can bypass ^E^FtsN unconditionally when mutated, as mentioned earlier. This would suggest that the CCD of FtsB would be a key suppressing region in the context of sPG synthesis. However, there are no indications that residue E56 interacts with either FtsL, FtsQ, FtsI or PBP1b, as would be expected if the region allosterically suppresses sPG synthesis [26]. Furthermore, the CCD region is not involved in FtsI or FtsL recruitment, as a truncated and prematurely targeted form of FtsB (ZapA-FtsB^1−55^) can still recruit both [112]. The structural prediction of FtsB does however indicate that the E56 stabilizes that distal part of the FtsB structure through backbone hydrogen bonds (Figure 13). Moreover, the CCD domain of FtsB is just upstream of a highly structured FtsQ-binding region (64–87) [98,100]. Together with the proposed position of the CCD domain by Craven et al. in the FtsBL tetramer, this suggests that the FtsB CCD domain indirectly provides stability to the FtsBL coiled coil structure, and thus delays sPG synthesis [111]. Overall, this indicates that increased stability in the FtsBL coiled coil suppresses sPG synthesis, while decreasing stability activates sPG synthesis.

#### 4.3.7. C-Terminal Interactions in the FtsBLQ Subcomplex Appear to Modulate the FtsBL Coiled Coil Stability

The inherent instability of the coiled coil domain seems to be stabilized by interactions in the C-terminal region of FtsBLQ. The helical structures of both FtsB and FtsL are indicated to break in or close to the CCD domains (Figure 10). Especially the peculiar conformation of the FtsL CCD domain is interesting, as it is predicted to break in the middle (Figure 13) [111]. E88 and N89 are at the end of the coiled coil, while G92, D93, and H94 are likely located in a loop. Interestingly, backbone hydrogen bonds between E88 and G92 in addition to N89 and D93 are predicted, which may stabilize the kink in the CCD domain of FtsL. The modelling of Craven et al. indicates that the break in the CCD may not occur in the presence of FtsQ, which indicates that the increased structural stability provided by FtsQ, combined with the formed backbone hydrogen bonds in the CCD, may regulate FtsBL conformation [111].

Furthermore, it is predicted that FtsBL also interacts past the CCD domain at the C-terminus (Figure 10 and Figure 15). The interaction partly overlaps with the FtsB–FtsQ-binding region, forming a multi-stranded β-sheet, which is proposed to stabilize the FtsBL region (Figure 15; βL, βB and βQ). The deletion of the FtsB region just downstream (86–103) results in the diminished recruitment of FtsN, while downstream recruitment is not affected [112]. Similarly, a C-terminal truncated form of FtsL can still interact with FtsB and recruit downstream proteins, but results in filamentous growth and diminished interaction with FtsQ [113]. This interaction is close to a FtsQ residue (S242) that is involved in FtsN binding, which suggest that the C-terminal FtsBL interaction may induce a FtsN-binding conformation of FtsQ [85]. The binding of FtsN to the C-terminal domain of FtsQ may induce instability in the FtsBL C-terminal interaction, which would subsequently affect the stability of the coiled coil domain (Figure 15; S242). Overall, a C-terminal FtsBLQ interaction appears to stabilize the FtsBL coiled coil, and is also involved in FtsN recruitment, which is proposed to indirectly induce instability in the coiled coil structure.

#### 4.3.8. The FtsBLQ Subcomplex Allosterically Controls sPG Synthesis through Suppressing and Activating Interactions, the Latter Induced by FtsN

Here, a working model is produced describing the regulation of the FtsWI–PBP1b by FtsBLQ, based on the discussed data described above (Figure 16). The rate-limiting step during sPG synthesis appears to be the activity of FtsWI [65,114,115]. The complex formation of FtsW and FtsI increases the activity of FtsW to a baseline level, which is then increased or suppressed by regulators [19,25,26]. The FtsWI complex appears to be inherently unstable in the periplasmic domains, which may also affect PBP1b activity [26,50]. Thus, FtsWI–PBP1b appears to have a low-activity level, which provides an excellent basis for regulators to act on to either suppress or increase FtsWI–PBP1b activity.

FtsBLQ stabilizes FtsWI–PBP1b by interacting with certain domains (Figure 16). Some of these regulating interactions appear to be excluding others, especially the activating FtsL–FtsI interaction and the suppressing FtsL–PBP1b interaction. This indicates different conformations of FtsBLQ that either promote or suppress sPG synthesis. The default conformation of FtsBLQ appears to stabilize FtsWI-PBP1b in a more inactive state after recruitment through a cytoplasmic FtsL–FtsW interaction (Figure 16a). The main interactions promoting the inactive state are proposed to be the FtsL CCD–PBP1b and FtsQ–FtsI interactions (Figure 16b,d). The inactive conformation of FtsBLQ is promoted by the increased stability of the FtsBL coiled coil structure, potentially through a C-terminal FtsBL interaction (Figure 16c) [98,100,111].

As earlier discussed, FtsN likely switches FtsBLQ to an active conformation. It has been suggested that FtsN may bind to multiple domains of FtsQ [85]. An interaction close to the inner membrane may lead to dissociation of the suppressing FtsQ–FtsI binding (Figure 16f). FtsN further binds to the C-terminal domain of FtsQ, which may destabilize a C-terminal FtsBL interaction, and thus induces coiled coil instability (Figure 16h). Inducing instability in the coiled coil appears to switch FtsBL to an active conformation, which promotes an activating interaction between the AWI domain of FtsL and the FtsI pedestal domain (Figure 16g). Furthermore, the FtsL CCD domain is proposed to be outcompeted by the ^E^FtsN domain for PBP1b, which leads to the accelerated activity of PBP1b (Figure 16e). This last interaction further promotes the FtsL AWI–FtsI interaction, as FtsL is not bound anymore to PBP1b. Overall, FtsBLQ stabilizes the inherent instable structure of FtsWI–PBP1b in an inactive conformation, which switches to an active state promoted by conformational changes induced by FtsN.

Interestingly, the essentiality of the small ^E^FtsN domain seems to be accounted for by the proposed model, as the absence of ^E^FtsN would negatively affect sPG synthesis in two manners. Firstly, the suppression of PBP1b activity by the FtsL CCD domain would not be alleviated. Secondly, it may lead to a stable sequestering of FtsL to PBP1b away from FtsWI, preventing the activating FtsL AWI–FtsI interaction. The prevention of the ^E^FtsN–PBP1b interaction does indeed seem to keep FtsWI–PBP1b in a more inactive state, as is indicated by the mild cell chaining and lysis caused by mutations in the ^E^FtsN-binding pocket of PBP1b [28]. Other FtsN domains may be less essential due to the nature of the allosteric interaction they make with the synthesis complex. Furthermore, this would also account for the nature of FtsB, FtsL and FtsA SF mutants. Except for FtsB^E56^, SF mutants need additional activating mutations in other regulators to bypass the necessity of FtsN [16]. They might also bypass an inactive ^E^FtsN domain, but not a complete absence of FtsN. This indicates that these mutants may promote stabilization of the synthesis machinery in an active conformation in cooperation with other mutants or with remaining FtsN domains. Indeed, the sum of multiple allosteric interactions between FtsBLQ, FtsN and FtsWI–PBP1b stabilizes the sPG synthesis machinery in inactive or active conformations, leading to the suppression or acceleration of septal peptidoglycan synthesis.

## 5. An Updated Model

From here, we can decipher a new model for septal peptidoglycan synthesis in *E*. *coli.* This model is conditional, as it assumes that FtsN is the only activating signal (not likely due to DedD), and is heavily based on mutagenesis studies, which need to be validated by direct protein–protein interactions, subcellular localization, and in vitro mode of actions.

In the model, the recruitment of the divisome is completed before the start of constriction. This includes the proto-ring (FtsZ, FtsA, ZipA), FtsEX, and a large synthesis complex consisting of FtsK, FtsBLQ, FtsWI, and PBP1b. FtsA is then still in an oligomerized state and thus delays the septum formation. Septation can start when FtsN recruits to the division site and FtsA converts to a monomeric conformation [81]. At this moment, septal peptidoglycan synthesis is activated. In the new model, FtsN activates septal peptidoglycan synthesis by stabilizing the FtsWI–PBP1b complex in an active conformation (Figure 16 and Figure 17). FtsN accumulation at midcell is likely accelerated by endopeptidase and amidase activity, which may lead to an increased number of synthesis complexes on the slowly moving synthesis track, away from the FtsZ assembly track (Figure 9) [27,29,84].

Based on the role of FtsK as a cell division DNA segregation checkpoint, we expect the whole synthesis machinery, including FtsK, to translocate. Unfortunately, only the dynamics of FtsN and FtsWI have been thoroughly examined for now [29]. If the hexamer FtsK is also on the active track, this would provide a scaffold for structured sPG synthesis. Earlier stoichiometry models proposed that each FtsK hexamer (~seven on average at midcell) was linked to four FtsWI, four PBP1b, four FtsBLQ, and eight FtsN molecules [116]. A compelling protein concentration analysis at midcell reported a different stoichiometry (FtsK hexamer surrounded by 7–9 FtsI, 2–3 PBP1b, 2–3 FtsBLQ, and ~15 FtsN molecules) [62,117]. This could be an underestimation for FtsBLQ and PBP1b due to antibody epitope recognition hinderance by complex formation, as ribosome profiling determined a similar copy number for FtsI, FtsQ, FtsB, and PBP1b in minimal medium [118]. However, given the reported hexamer conformation of FtsQ at midcell, each FtsK molecule likely interacts with one FtsQ molecule [117]. Such a synthesis complex is proposed to consist of three synthesis nodes, each containing two FtsBLQ linked to two FtsWI–PBP1b subcomplexes, mostly based on the tetrameric nature of FtsBL and isolated FtsBLQ–FtsWI–PBP1b [15,26,110] (Figure 17). This model still needs to be validated (see the underestimation for FtsI and FtsN). However, the low number of complexes (10 at most) and large number of subunits (at least 26), combined with its potential mechanism of inward PG synthesis, makes it a great candidate to be validated with single-molecule localization microscopy techniques, as has been performed on bacterial protein secretion systems [119].

Interestingly, peptidoglycan in *E. coli* is predicted to be mostly single-layered based on atomic-scale simulations, with around 25% being three-layered, equivalent to the surface of the cell poles [120,121,122] (Figure 18). This would be in line with three synthesis nodes, where each node makes up two layers, each consisting of two NAM-NAG strands. The synthesized strands are crosslinked and then divided by PG hydrolases in two separate PG meshes consisting of three PG layers. PG strands at both the poles are likely long (up to 200 nm) and in a circumferentially ordered orientation, similarly to the conformation of PG at the cylindrical part of the cell, which would be consistent with multiple synthesis complexes travelling (bi)directionally over the division septum [95,122]. From the available data, we deduced a theoretical model where each synthesis complex synthesizes a PG mesh of six strands wide and two strands high (Figure 18). The PG mesh is split in two three-strand-wide and two-strand-high meshes by (lagging) PG hydrolases, one for each future daughter cell. Thus, the proposed stoichiometry of the synthesis complex and corresponding septal peptidoglycan production provide a model of structured remodeling where the integrity of the peptidoglycan is not compromised.

### Concluding Remarks

This model is a starting point and thus still very speculative. Most of the review is focused on the direct regulation of the septal peptidoglycan synthesis machinery through allosteric interactions with regulators. In this review, the combination of protein–protein interaction predictions and extensive literature research uncovered several putative interactions within FtsWI and with regulators that may stabilize the synthesis complex in active or inactive conformations. The next step is the validation of these interactions, as this would elucidate their nature. It is a fascinating model, based on a balance of multiple positive and negative allosteric interactions, of which the sum leads to the activation or inhibition of septal peptidoglycan synthesis.

Validation of the proposed interactions is needed, as a large part of the genetic analyses conducted with the DN and SF mutants of divisome proteins are heavily reliant on spot assays. These analyses rely on the principle that the (over)expression of certain mutants, or the overexpression/deletion/depletion of certain proteins, may result in varying bacterial growth and viability. These methods provide a quick and straightforward determination of growth and viability in response to mutagenic changes. However, these spot assays, which are in essence ‘blackbox system’ experiments, lack validation of interactions and the assessment of FtsWI–PBP1b activity. Fortunately, in vitro activity and interaction assays have been conducted in the past. The most detailed interactions are those between PBP1b and ^E^FtsN, and FtsB and FtsQ [28,98,105]. Especially, the FtsL CCD–PBP1b and FtsL AWI–FtsI interactions lack validation and are mostly supported by genetic analyses and spot assays [23,26]. Similar analyses, as performed for the ^E^FtsN–PBP1b interaction, could confirm the abovementioned interactions, which are key parts of the proposed model.

Furthermore, not many in vitro studies have been published in the last few years. The main papers concerning the in vitro effect of FtsBLQ/FtsN on FtsWI–PBP1b activity are the papers of Boes et al., and Marmont and Bernhardt [25,26]. The differences in procedure between the two studies make definite interpretation difficult. However, these studies served as guidance because in vitro data concerning the discussed subject is limited. Fortunately, these studies provide methods to assess the effect of certain FtsBLQ/FtsN domains on the catalytic activity of FtsWI and PBP1b. Especially the fluorescent dansyl lipid II assay of Boes et al., which can be performed with functional FtsI, is a major step forward in this line of research. It has recently been employed in a study of the effect of ^E^FtsN on PBP1b GTase activity, and will likely be used in future studies of other activating or suppressing interactions [28].

Interesting years lie ahead, as this is only a glimpse into the world of the regulation of bacterial cell division. So, what to think about the role of peptidoglycan hydrolases, especially linked to the semi-essential FtsEX subcomplex? While only briefly discussed in this review, the activity of these proteins seems to contribute to the spatial distribution of sPG synthesis (Figure 9). Though still theoretical, this would coordinate PG synthesis with hydrolysis, leading to optimal septal formation and daughter cell separation. Furthermore, the fact that only the PBP1b-activating domain ^E^FtsN is essential for cell division, combined with the partial redundancy between FtsN and DedD, indeed indicates alternative activation factors. DedD was not included in the model, as the direct position of DedD in the divisome is not yet discovered. This will most likely happen in the coming years, as DedD might represent another activation pathway, parallel to FtsN and perhaps through FtsWI. Luckily, these additional factors can easily be incorporated in the proposed model, as these would further contribute to an inactive or actively synthesizing state of FtsWI–PBP1b.

Overall, an interesting model is provided that integrates multiple aspects of septal peptidoglycan synthesis. This includes a spatial model for the synthesis complex, multiple regulatory mechanisms, and a compelling organization model for newly synthesized PG strands. While still a pipe dream for now, the model provides new approaches for the development of new antibiotic compounds. Consider destabilizing the FtsWI regulatory site with small-molecule inhibitors, and the development of a compound that binds to the FtsL AWI domain or prevents synthesis complex recruitment with compounds mimicking the FtsQ domain. Altogether, the complexity of septal peptidoglycan synthesis is challenging, but simultaneously provides many opportunities for the development of new antibiotic compounds. This may provide us with new tools to tackle the emergence of antibiotic resistance, which is one of most significant challenges humanity will face in the coming few decades.

## Figures and Tables

**Figure 1 ijms-23-03537-f001:**
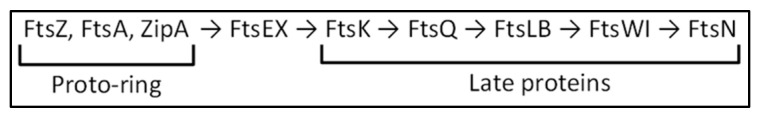
Hierarchical recruitment of divisome proteins.

**Figure 2 ijms-23-03537-f002:**
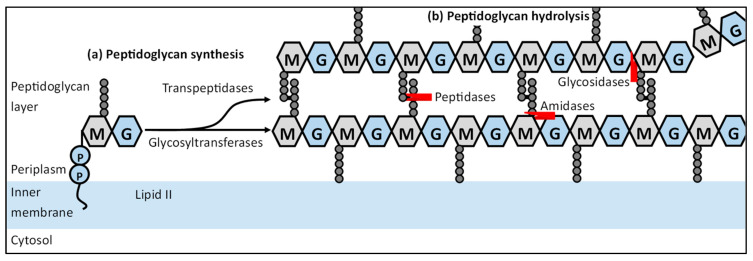
Peptidoglycan biosynthesis and remodeling. This simplified scheme describes the steps needed to synthesize new peptidoglycan from precursor Lipid II that is synthesized in the cytoplasm. Lipid II is then flipped to the periplasmic side of the inner membrane by a flippase, where the synthesis of peptidoglycan occurs. (**a**) Lipid II is incorporated in an already growing peptidoglycan strand by glycosyltransferases. Adjacent Peptidoglycan strands are crosslinked by transpeptidases through the pentapeptides attached to the NAM residues, resulting in a peptidoglycan mesh. (**b**) Factors involved in the remodeling of the peptidoglycan mesh are PG hydrolases, which can be divided in multiple groups based on their targets. Peptidases cleave amide bonds in the cross-linked peptide chains, amidases cleave off the peptide chains attached to NAM residues and glycosidases that hydrolyze the glycosidic linkage between adjacent NAG and NAM residues.

**Figure 3 ijms-23-03537-f003:**
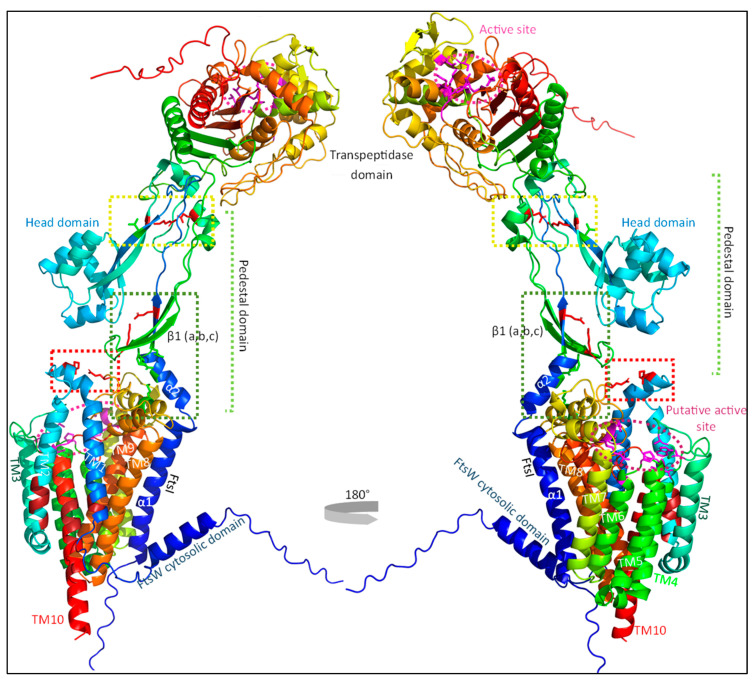
The FtsWI peptidoglycan synthesizing holoenzyme. FtsI consists of a short cytoplasmic domain, a transmembrane domain (α1), a pedestal domain (including head domain) and a C-terminal transpeptidase domain. FtsW comprises a collection of 10 transmembrane domains (TM1-10) which are connected by periplasmic and cytoplasmic loops. Multiple regions that control the activity of FtsWI are indicated by the colored boxes and detailed further in Figure 3 and Figure 4. Although no crystal structure of the FtsWI is available, predictions of the structure have been produced with AlphaFold2 advanced and visualized with the PyMOL Molecular Graphics System, Version 2.5.2 Schrödinger, LLC. Sequences for the prediction were obtained from the UniProtKB database (P0ABG4-1 for FtsW and P0AD68-1 for FtsI).

**Figure 4 ijms-23-03537-f004:**
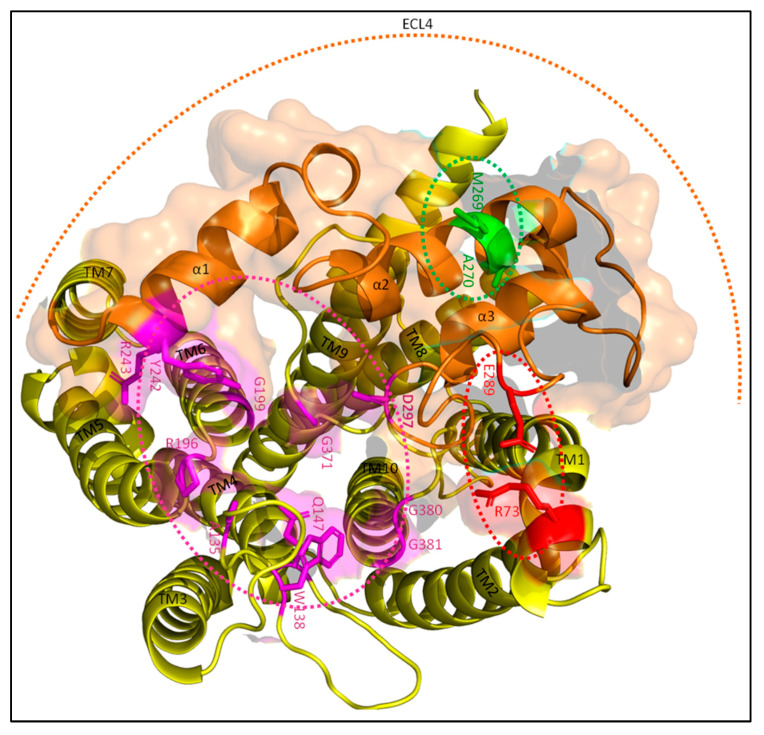
FtsW active site. A single FtsW protein (yellow) is shown from above in the periplasm. The active site is indicated by the magenta–dotted ellipse with active site residues also highlighted in magenta. The ECL4 (orange) contains the putative catalytic residue D297, a region around M269 (green) that activates FtsW, and another region around E289 (red) that inhibits FtsW. E289 forms a putative interaction with R73 in this prediction. α1, α2 and α3 indicate the three α-helixes of ECL4. The FtsI interaction interface is highlighted in transparent orange. The active site residues are from the paper of Li et al. [67].

**Figure 5 ijms-23-03537-f005:**
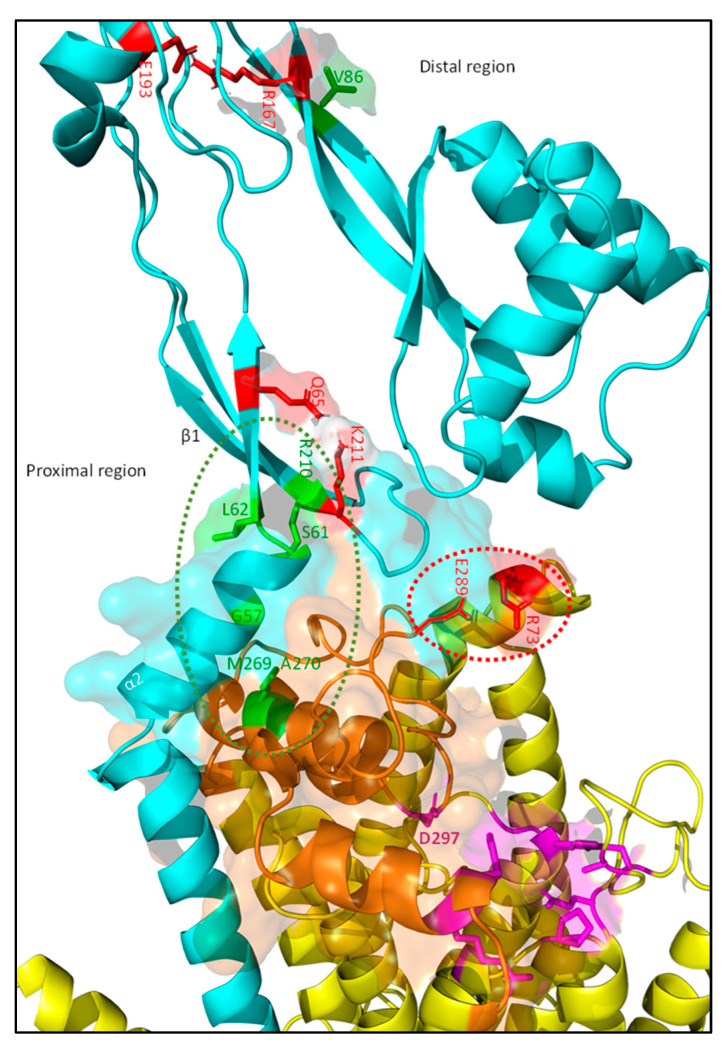
FtsI pedestal domain interaction with FtsW ECL4. FtsI (cyan) and FtsW (yellow) interact in the periplasm leading to the activation of FtsW. The membrane proximal part of the FtsI pedestal domain interacts with ECL4 of FtsW (orange). Activating (green) and suppressing (red) residues can be found in this region, and likely lead to stabilization of the active residue FtsW^D297^ in the active site. The proximal part of the pedestal domain contains multiple residues divided over an α-helix and β-sheet that activate FtsW. These include residues G57, S61 and L62 located on α2, and residue R210 on β1. Residue K211, located adjacent to R210, inhibits FtsW activation, likely through an interaction with Q65. The pedestal domain interacts mostly with the FtsW region around M269, which activates GTase activity. A cluster of residues at the end of the pedestal domain is important for the stability of this domain and appears to be involved in the regulation of the FtsI-mediated activation of FtsW. Residue R167, proposed to interact with E193, stabilizes the domain and thus inhibits FtsW activation, though it may also affect the activity of the FtsI TPase domain.

**Figure 6 ijms-23-03537-f006:**
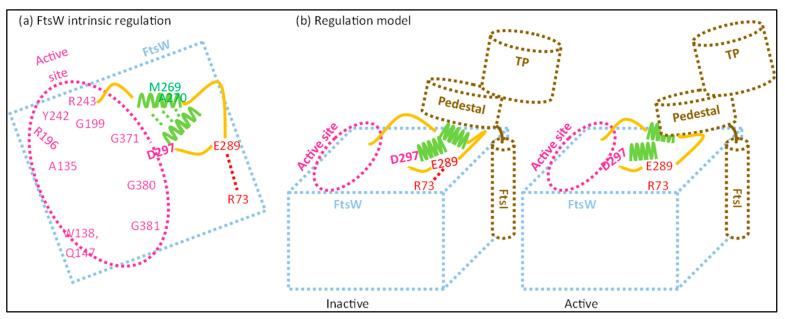
Intrinsic FtsWI regulation. FtsWI appear to have an intrinsic regulatory mechanism, which is expected to be affected by regulators. In a simplified scheme of the periplasmic side of FtsW (**a**), the catalytic residue D297 is located at the periphery of the active side (purple–dotted ring) in the loop between TM7 and TM8 that also contains the inhibiting E289 residue and the activating M269 and A270 residues. A balance between these two regions on the same loop appear to regulate FtsW GTase activity. A putative interaction between E289 and R73, located on TM1, likely destabilizes or sequesters D297 from the rest of the active site, leading to an inactive conformation. The hydrophobic region around M269 and A270 may antagonize this and lead to the stable positioning of D297 in the active site. (**b**) A speculative model describing the possible activation of FtsW by FtsI is shown. Regions of the pedestal domain of FtsI may interact with the α-helix containing M269 and A270, pushing the regulatory loop in an active conformation.

**Figure 7 ijms-23-03537-f007:**
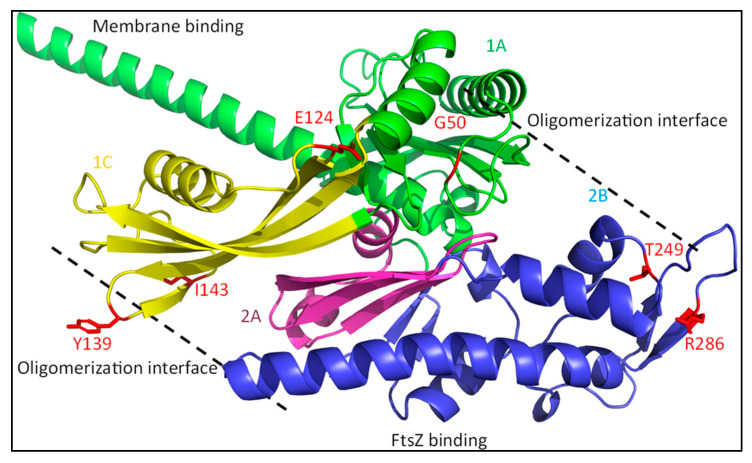
FtsA structure. FtsA in a cytoplasmic protein that binds to the inner membrane and acts as a membrane anchor for FtsZ. FtsA is divided into four domains which interact with different divisome components [81]. The 1A domain (green) is involved in membrane binding and has also been linked to FtsK binding (see text). The 2B domain (blue) contains a region that binds FtsZ and, together with the 1A and 2A domains, forms a nucleotide (ATP)-binding pocket at the FtsA core. The 1C domain (yellow) is involved in the recruitment of FtsN. FtsA oligomerizes through 1A–1C and 2A–2B interactions. Multiple SF mutants are shown in red, which are mostly located at the oligomerization interfaces.

**Figure 8 ijms-23-03537-f008:**
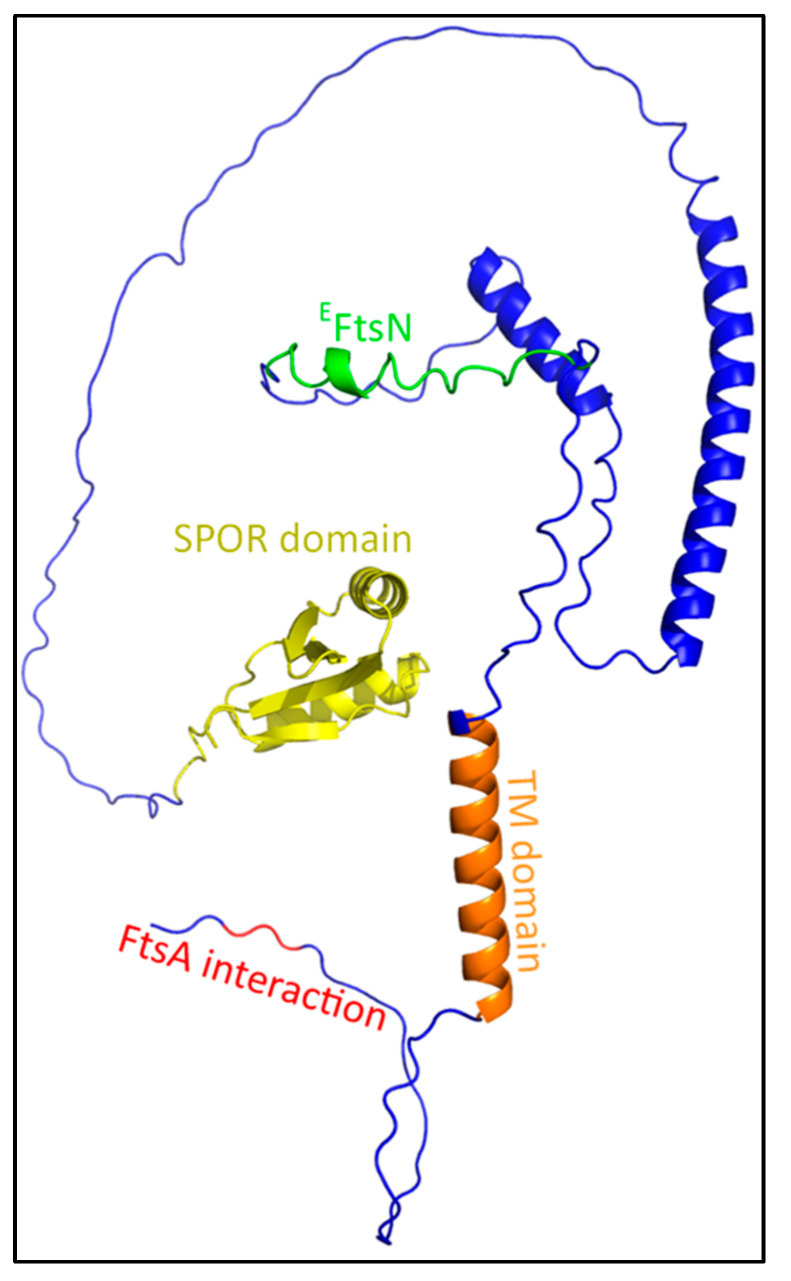
FtsN structure. FtsN is a long bitopic protein, with a small cytoplasmic domain that interacts with FtsA (red), a transmembrane domain (orange), and a large periplasmic domain. The periplasmic domain is mostly unstructured, ending in a SPOR domain that binds to denuded glycans. The region in between the membrane and the SPOR domain contains three α-helixes, including an essential domain of 18 residues (green: ^E^FtsN domain) partly located on the first helix.

**Figure 9 ijms-23-03537-f009:**
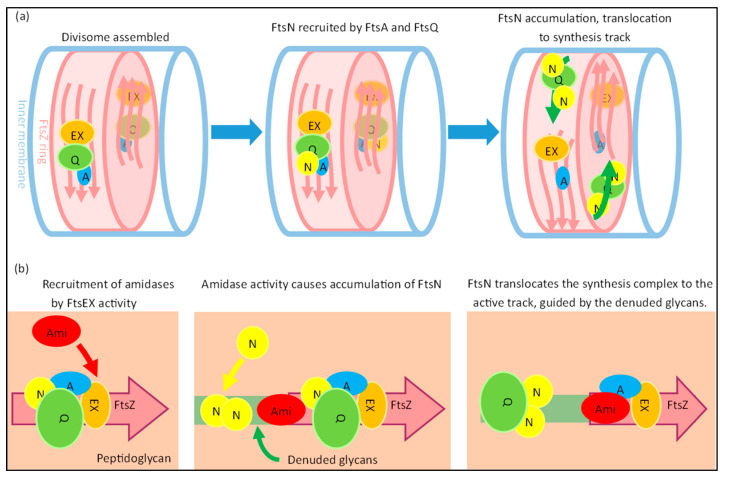
FtsN active track formation. (**a**) A simplified model describing the possible formation of the FtsN-associated active track is shown. When the divisome is completely assembled, FtsN (N; yellow) can be recruited to the division site. FtsN is likely recruited by direct interactions with FtsA (A; blue) and FtsQ (Q; green). FtsEX (EX; orange) ATPase activity of FtsEX leads to the recruitment of amidases (Ami; red), which form denuded glycans. The SPOR domain of FtsN binds to the denuded glycans, leading to the accumulation of FtsN at the septum. FtsN, together with the synthesis complex represented by FtsQ, then translocates to the slower, active track away from the faster FtsZ-associated track. (**b**) The active track is proposed to be ‘carved’ out in the septal peptidoglycan by the amidases activated by FtsEX, which tightly interact with FtsZ from its recruitment. This mechanism describes how the presumed translocation of the synthesis complex may occur. This links the FtsN-track and FtsZ-track spatially, in line with experimental data. The pink arrows indicate the dynamics of FtsZ filament bundles, and the green arrows at the top indicate the dynamics of the active FtsN-associated track.

**Figure 10 ijms-23-03537-f010:**
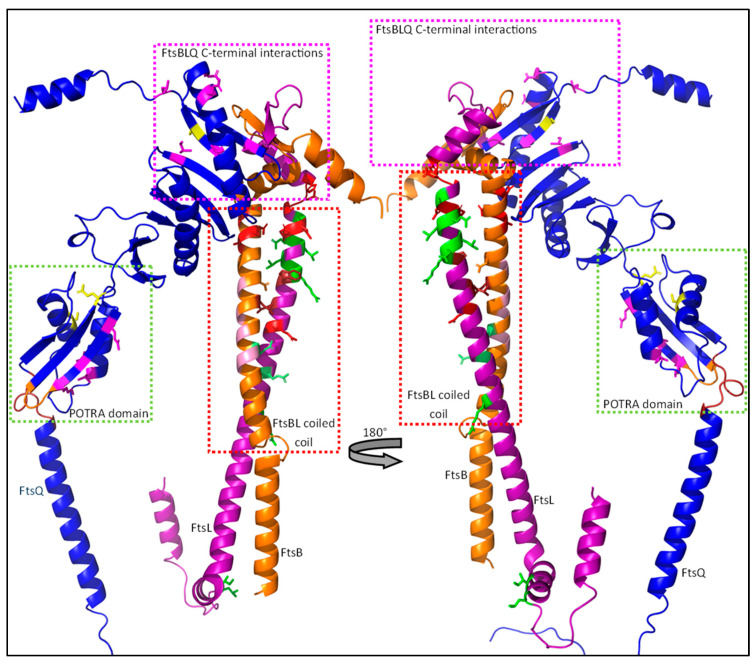
The FtsBLQ regulatory subcomplex. Shown is the FtsBLQ subcomplex responsible for the regulation of the septal peptidoglycan synthesis machinery. The largest subunit, FtsQ, acts as a scaffold and comprises a small cytosolic domain, a transmembrane domain, and a large periplasmic domain. FtsB and FtsL have a similar structure, mostly consisting of a long helical structure, and form a coiled coil together. FtsB has a short cytoplasmic tail, while FtsL carries a large cytosolic double helical structure involved in the recruitment of FtsW. Regions or interactions that are important for sPG synthesis regulation are highlighted by the colored boxes and are further detailed in Figure 11 (POTRA domain), Figure 13 (FtsBL coiled coil) and Figure 15 (FtsBLQ C-terminal interactions). The structures shown here are predictions produced with AlphaFold2, advanced and visualized with the PyMOL Molecular Graphics System, Version 2.5.2 Schrödinger, LLC. Sequences for the prediction were obtained from the UniProtKB database (P06136-1 for FtsQ, Q9HXZ6-1 for FtsB and P0AEN4- for FtsL).

**Figure 11 ijms-23-03537-f011:**
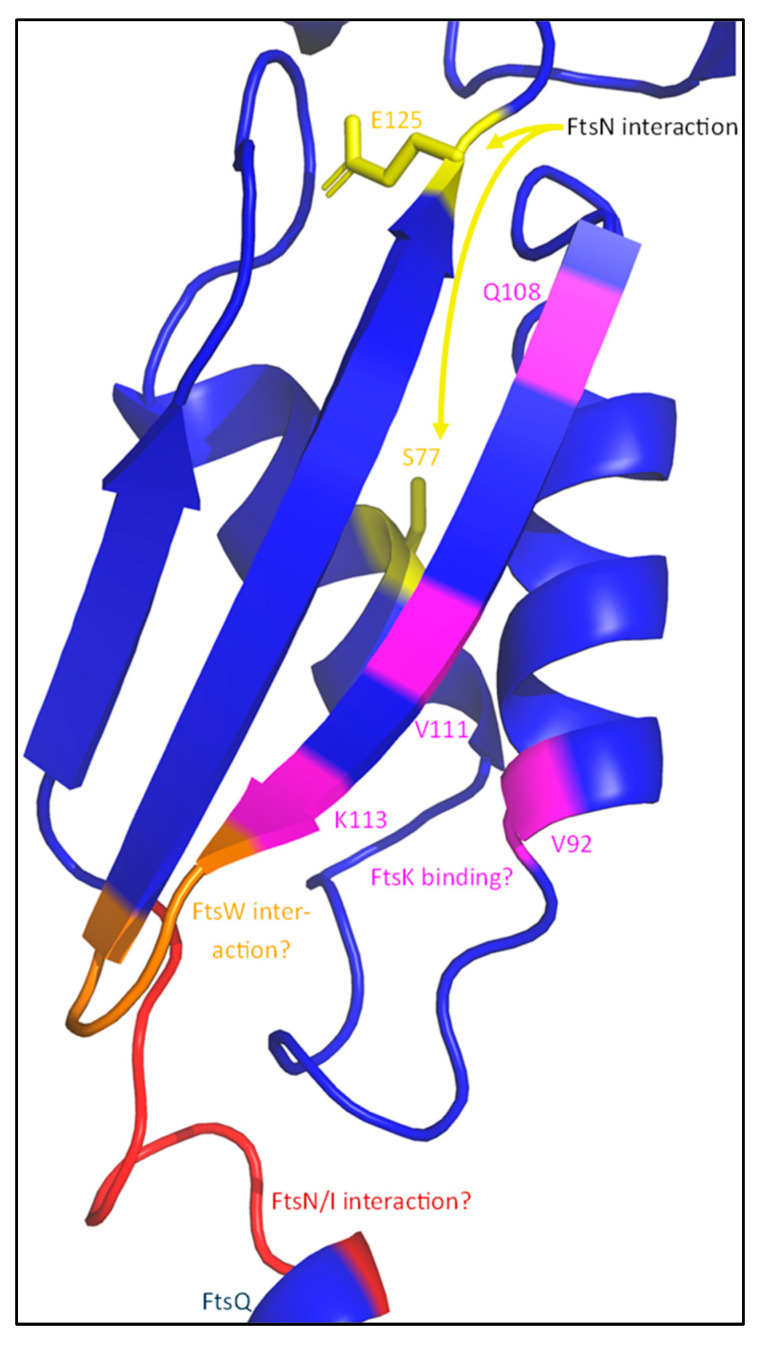
The FtsQ POTRA domain. FtsQ interacts with multiple divisome proteins through its POTRA domain, which is part of the membrane-proximal periplasmic domain. A loop between the TM domain and the first strand of the β-sheet (50–57 in red) is necessary for the interaction with FtsI (and FtsN). Additional FtsN interacting residues are shown in yellow (S77, E125). The POTRA domain also contains a FtsK-binding site (purple), which includes a hydrophobic pocket (V92, V111, L57). Furthermore, a FtsW interaction interface is expected somewhere in the region between Q108 and E125. This is likely the loop between the two strands of the β-sheet, as this is the region predicted to be closest to the membrane-embedded FtsW (orange).

**Figure 12 ijms-23-03537-f012:**
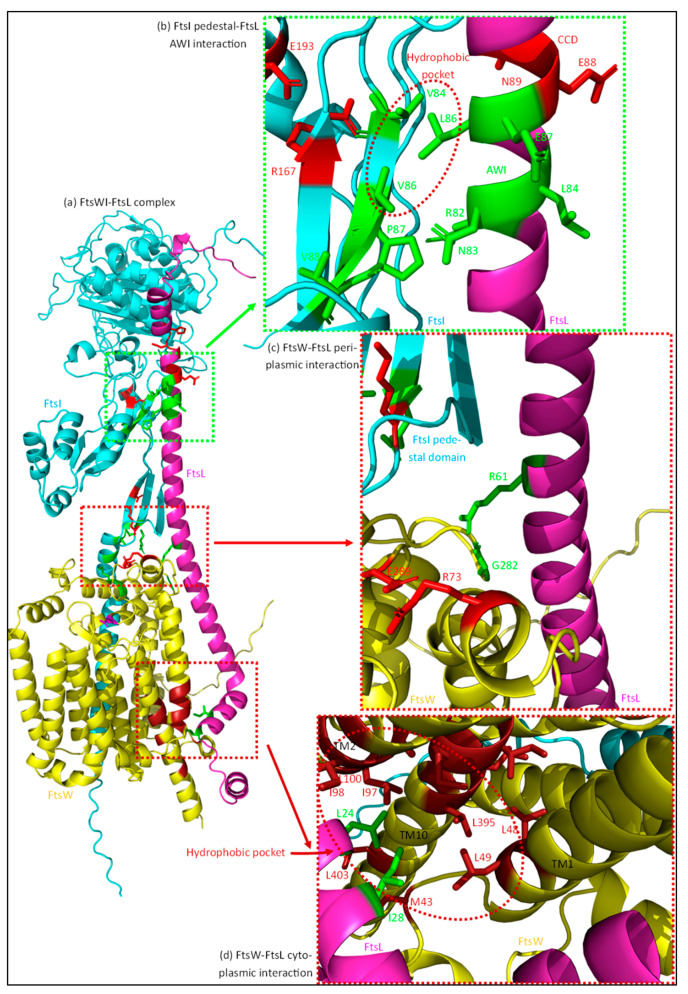
Interactions between FtsL and FtsWI. FtsL interacts with multiple regions of FtsWI, which likely leads to a stable active conformation. (**a**) The complex formed between FtsW (yellow), FtsI (cyan) and FtsL (purple) is shown. (**b**) A periplasmic interaction between the FtsL AWI domain (green) and the pedestal domain of FtsI is predicted to be important for FtsWI activation. A hydrophobic pocket around FtsI residues V84 and V86 (red dotted ellipse) appears to interact with FtsL residue L86. (**c**) FtsL also appears to interact with FtsW close to the IM on the periplasmic side. A hydrogen bond between FtsL residue R61 and FtsW residue G282 is formed in the predicted structure. The R61–G282 interaction (green) may stabilize the periplasmic loop in a more active conformation. (**d**) FtsL recruits FtsW through its cytoplasmic domain and this interaction also plays a minor role in FtsW activation. A hydrophobic pocket formed by residues om TM2 and TM10 of FtsW (brick-red) interacts with the FtsL cytoplasmic domain (L24, I28; green), possibly stabilizing the TM domains that carry active site-residues at the periplasmic side. The structures shown here are predictions produced with AlphaFold2 advanced, and visualized with PyMOL Molecular Graphics System, Version 2.5.2 Schrödinger, LLC. Sequences for the prediction were obtained from the UniProtKB database (P0AEN4-1 for FtsL, P0ABG4-1 for FtsW, P0AD68-1 for FtsI).

**Figure 13 ijms-23-03537-f013:**
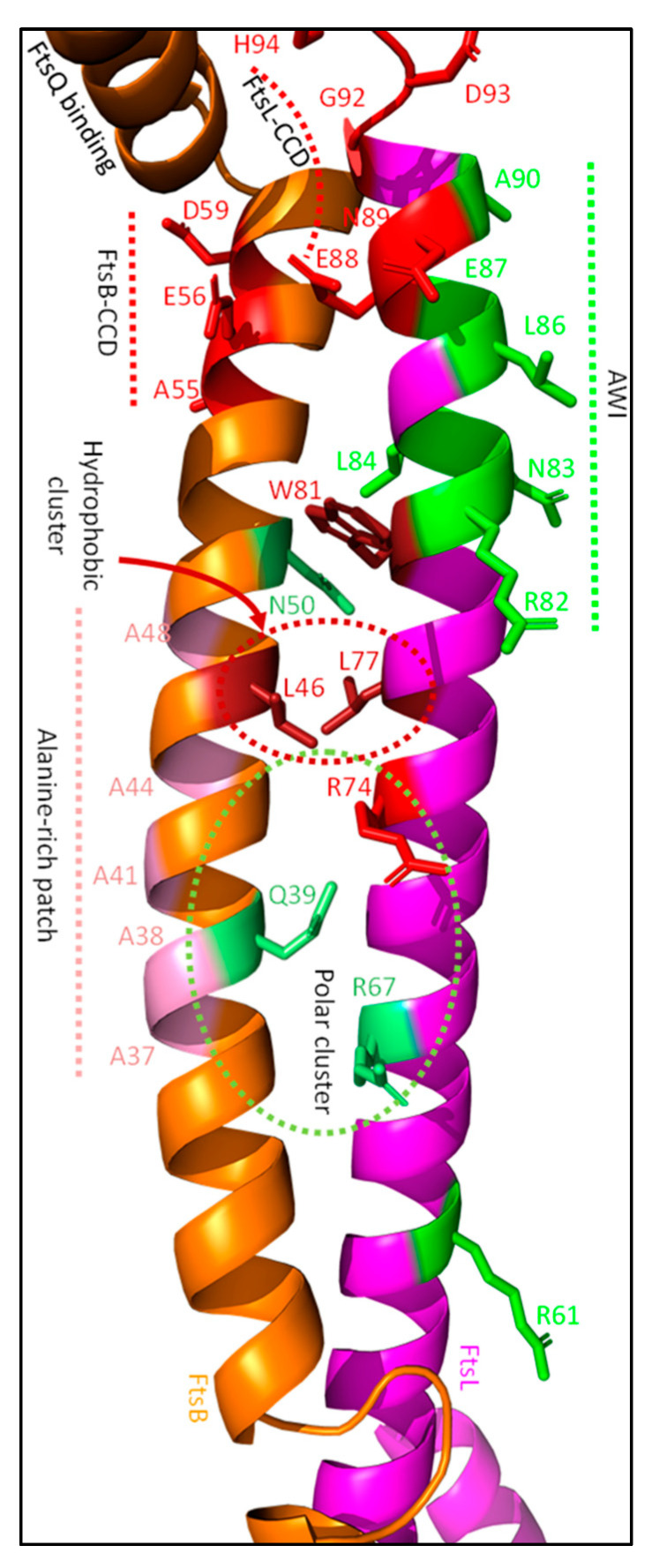
FtsBL coiled coil stability. Modulating the stability of the FtsBL coiled coil structure is an integral part of sPG synthesis regulation. The stability is dynamically affected by polar (Q39, R67, R74; N50) and hydrophobic (L46, L77, W81) clusters located at the inner part of the coiled coil between the transmembrane and the CCD/AWI domains. FtsB contains an alanine-rich patch at the other face of the helix, forming a dimerization/intercoil domain. The CCD (red) and AWI (green) domains just upstream are involved in the suppression and activation of sPG synthesis, respectively. Just above the FtsB CCD domain, a region stably interacting with FtsQ can be found.

**Figure 14 ijms-23-03537-f014:**
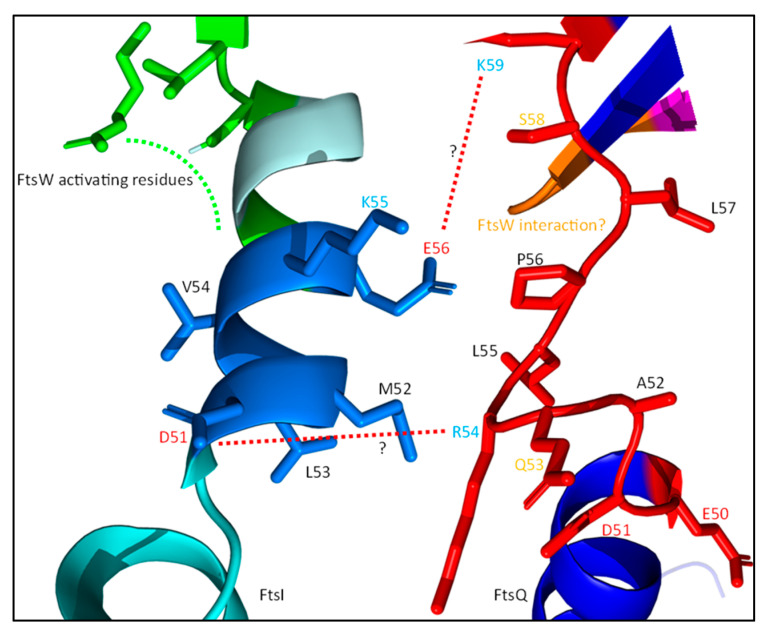
Potential FtsQ–FtsI interactions. The potential FtsQ–FtsI interaction interface described in Table 2 is shown here in the predicted structures of FtsI and FtsQ. The FtsI region forms a helix, while the FtsQ region is an unstructured loop. Possible regulating interactions have been indicated with the red lines and the FtsW-activating residues of FtsI in green.

**Figure 15 ijms-23-03537-f015:**
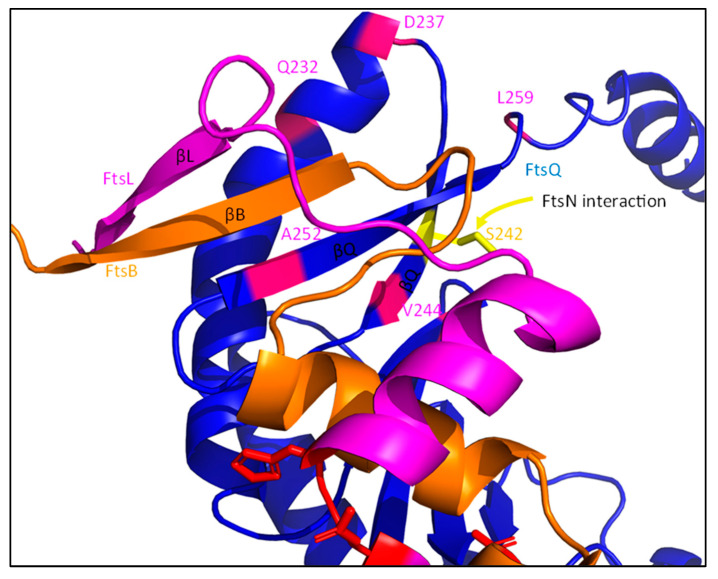
FtsBLQ C-terminal interaction. The C-terminal domain of FtsQ interacts with FtsB, through the formation of a multi-stranded β-sheet (βL, βB and βQ). Stability in this domain is important for the interactions, as mutating the residues marked in purple has a negative effect on FtsQ–FtsB interaction. Furthermore, a C-terminal interaction between FtsB and FtsL is predicted, as FtsL forms a β-strand interacting with the FtsB–FtsQ β-sheet (βL). This may be stabilized further by FtsQ. An additional FtsN-binding residue can be found in this region of FtsQ (S242; yellow) which is vital for FtsN–FtsQ interaction. This FtsN interaction may induce instability in this region, thus changing the conformation of FtsBL to an activation one.

**Figure 16 ijms-23-03537-f016:**
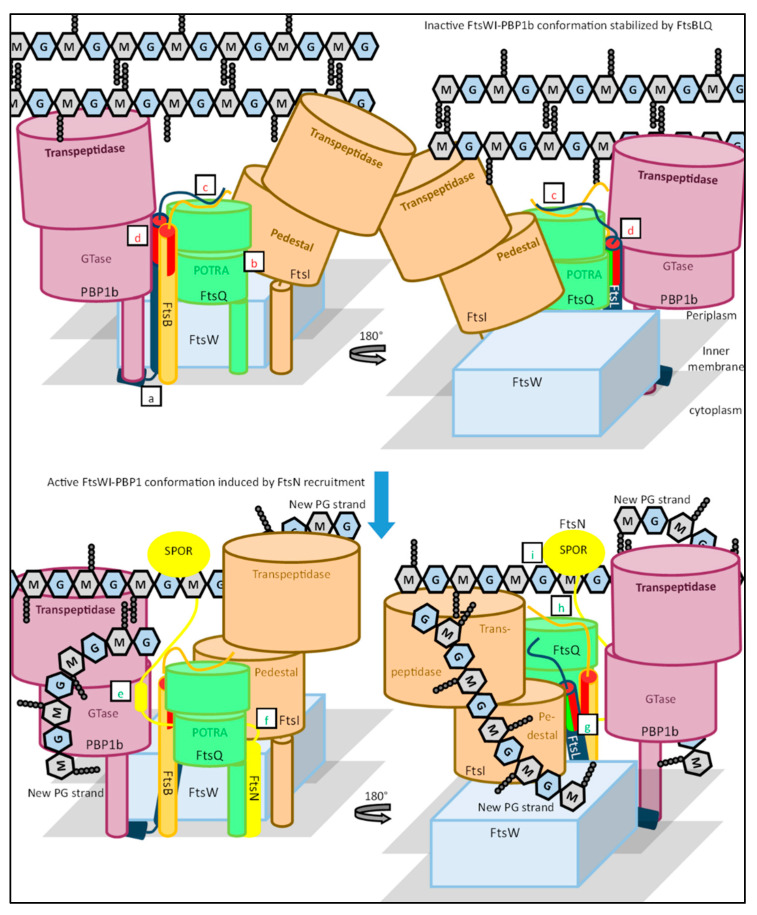
Regulation of FtsWI–PBP1b by FtsBLQ and FtsN. A model is proposed where FtsBLQ and FtsN stabilize the FtsWI–PBP1b septal peptidoglycan synthesis machinery in active or inactive conformations based on certain allosteric interactions. These interactions are numbered and either suppress (red), increase (green), or are neutral (black) regarding sPG synthesis. (**a**) FtsW/L cytoplasmic interaction. This interaction between FtsL and FtsW in the cytoplasm recruits the synthesis machinery to the division site. While primarily involved in recruitment, the interaction may also be involved in the stabilization of FtsW in a more active conformation. (**b**) FtsQ–FtsWI. The membrane proximal part of the FtsQ POTRA domain seems to interact with a periplasmic loop of FtsW (between TM1 and TM2) and the pedestal domain of FtsI. (**c**) C-terminal FtsBLQ interaction. A strong periplasmic interaction between FtsB and FtsQ is important for the stability of the FtsBL coiled coil structure, as well as an additional C-terminal FtsB–FtsL interaction that occurs in the same region. This increased stability is thought to stabilize the whole complex in an inactive conformation, thus negatively affecting sPG synthesis. (**d**) PBP1b–FtsL. A direct interaction between PBP1b and the CCD domain of FtsL (red) suppresses PBP1b GTase activity. (**e**) PBP1b–FtsN. FtsN (yellow) also directly interacts with the same PBP1b region, outcompetes FtsL–CCD, and leads to the activation of PBP1b GTase activity. (**f**) FtsN–FtsQ. FtsN is proposed to interact with the same FtsQ region as FtsI and leads to dissociation of the inhibitory FtsQ–FtsWI interaction. (**g**) FtsL–FtsI. An interaction between the now available FtsL–AWI domain (green) and the pedestal domain of FtsI stabilizes FtsWI in a highly active conformation, starting septal peptidoglycan synthesis. (**h**) C-terminal FtsBLQ (in)stability. An interaction between FtsN and the C-terminal domain of FtsQ (around residue S242) is proposed to lead to instability in the C-terminal FtsBLQ interaction through a conformational change in FtsQ. Together with interactions (**e**,**f**), the instability in the FtsBL coiled coil domain is increased, which leads to dissociation of the upper part of the coiled coil and subsequently to the activating FtsL–FtsI interaction (see interaction (**g**)). (**i**) FtsN SPOR interacting with denuded glycans. The SPOR domain of FtsN binds to denuded glycan as shown, and most likely directs the synthesis complex to ‘open’ PG, where synthesis can occur. New PG strands are formed when the whole synthesis complex is in its active state, induced by the sum of activating interactions (green). This model is not on scale and does not represent the possible stoichiometry of a complete synthesis node. M and G represent NAM and NAG residues, respectively.

**Figure 17 ijms-23-03537-f017:**
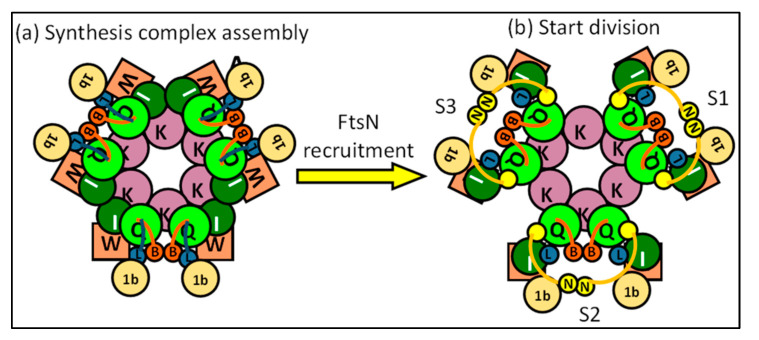
Stoichiometry of the synthesis complex. (**a**) A synthesis complex, shown from above in the periplasm, is assembled around FtsK hexamers, consisting of three synthesis nodes. Each synthesis node is comprised of 2 FtsBLQ–FtsWI–PBP1b subcomplexes, which assemble around FtsK through FtsQ. (**b**) FtsN recruitment activates the synthesis complex, leading to sPG synthesis by the three synthesis nodes (S1, S2 and S3).

**Figure 18 ijms-23-03537-f018:**
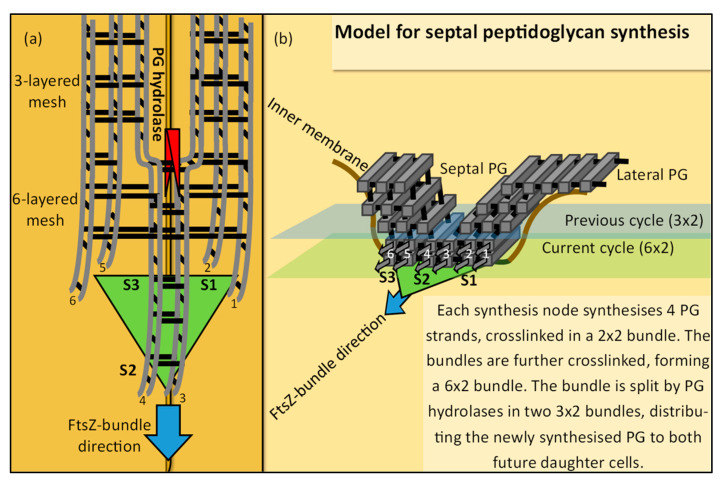
A speculative model describing peptidoglycan synthesis based on the synthesis complex. (**a**) At top-down scheme from the periplasm is shown. Here, the synthesis complex is in green and travels around the division site guided by FtsZ filament bundles (blue arrow). The three synthesis nodes (S1, S2, and S3) each produce a PG bundle consisting of 4 PG strands in a 2 × 2 conformation (grey lines = NAM-NAG polymers; black lines = crosslinked peptides). The numbers (1, 2, 3, 4, 5, and 6) indicate the two PG strands, vertically crosslinked in the model, which are further crosslinked with two PG strands synthesized by the other FtsWI–PBP1b subcomplex localized in the synthesis node. Further crosslinking of the newly synthesized PG bundles leads to a PG mesh of 6 strands wide and 2 strands high (6-layered mesh). The 6-strand-wide PG mesh is speculated to be divided by lagging PG hydrolases in two separate meshes, each 3 strands wide and 2 strands high (3-layered mesh). The two 3 × 2 PG meshes would be divided over two future daughter cells by the outer membrane, which would be in line with the current view concerning the structural conformation of PG at the cell poles. (**b**) We see the same speculative model describing septal peptidoglycan synthesis from a frontal view. Here, sPG synthesis is shown from a frontal perspective. The inner membrane (brown line) and the PG layer (grey blocks = NAM-NAG polymers; black lines = crosslinked peptides) are shown, while the outer membrane is excluded for simplicity. A distinction is made between lateral PG, which is mostly single-layered, while septal PG is three-layered in the model. The synthesis complex (green) produces the 6 × 2 PG mesh, which is not yet separated by PG hydrolases (not shown here) in the current cycle of remodeling. The PG mesh produced in the previous cycle has already been separated in two 3 × 2 meshes, where each mesh is distributed over a future daughter cell. This model, based on the supposed conformation of the synthesis complex, provides a cycle-based mechanism, where during each cycle a new PG mesh is produced inward from the PG produced during the previous cycle (or from a lateral, single-layered PG). The separation by PG hydrolases during the previous cycle may lead to denuded glycans, which can be bound by FtsN, providing a platform for the synthesis complex to produce new PG in an inward manner.

**Table 1 ijms-23-03537-t001:** ^E^FtsN and FtsL AWI/CCD domains.

Region				Residues Located in the PBP1b-Binding Pocket														
**^E^FtsN**								−		−	+			−	+			
	**P**	**P**	** K **	**P**	**E**	**E**	** R **	**W**	** R **	**Y**	**I**	** K **	E	L	E	S	R	Q
								**L**		**W**				S				
								**T**		**S**								
	**76**	**77**	**78**	**79**	**80**	**81**	**82**	**83**	**84**	**85**	**86**	**87**	88	89	90	91	92	93
**FtsL AWI/CCD**	I	E	W	R	N	L	I	L	E	E	N	A	L	G	D	H	S	R
				E	K	K		F	K	K	S	E		D	G	Y		
										V								
	79	80	81	82	83	84	85	86	87	88	89	90	91	92	93	94	95	96
									*i*	*i*	*i*	*i*			*i*	*i*		

The sequences of the essential domain of FtsN (^E^FtsN) and the FtsL AWI/CCD domains are shown. The amino acids are colored based on their physiochemical properties (red = negatively charged; blue = positively charged; yellow = polar; black = non-polar; grey = non-polar aromatic). The FtsN residues that reside in the PBP1b-binding pocket are highlighted in pale green, the FtsL CCD domain in pink, and the AWI domain in green. FtsN residues that directly interact with the binding pocket are indicated with a +, while those indicated with-form a hydrophobic pocket in the binding domain that is important for activation. Under the main sequence, amino acid substitutions are shown that negatively affect the activity of the FtsN domain. Similarly, the substitutions in the AWI domain (green) also negatively affect sPG synthesis, while the CCD domain substitutions have a positive effect. The columns indicated with *i* show the location of residues important for both FtsN and FtsL functioning. Most of the data concerning ^E^FtsN mutations are from Boes et al. (2020) and Liu et al. (2015), while the FtsL CCD/AWI mutants can be found in the main text.

**Table 2 ijms-23-03537-t002:** FtsQ–FtsI interaction region.

					Possible Interaction Interface																
FtsQ	E 50	D 51	A 52	Q 53	R 54	L 55	P 56	L 57	S 58	K 59	L 60	V 61	L 62	W 63	G 64	E 65	R 66	H 67	T 68	W 69	R 70
FtsI	V 47	I 48	S 49	P 50	D 51	M 52	L 53	V 54	K 55	E 56	G 57	D 58	M 59	R 60	S 61	L 62	R 63	V 64	Q 65	Q 66	V 67

The sequences of FtsQ (50–70) and FtsI (47–67) are shown. The amino acids are colored based on their physiochemical properties (red = negatively charged; blue = positively charged; yellow = polar; black = non-polar; and grey = non-polar aromatic). The possible interaction is highlighted in dark blue, while FtsI residues that are involved in FtsW activation are highlighted in green.

## Data Availability

Not applicable.

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
