# Peer review of "An Updated Model of the Divisome: Regulation of the Septal Peptidoglycan Synthesis Machinery by the Divisome"

_ijms, 2022, doi:10.3390/ijms23073537_

Round 1

Reviewer 1 Report

The manuscript is a review of septal peptidoglycan synthesis, an essential process required to separate daughter cells during bacterial cell division. This review was prompted by a series of recent findings from E. coli that necessitate an updated model to account for the current literature. The review itself is composed of five sections focused on background (sections 1 – 3), regulation (section 4), and a final model (section 5). The authors’ discussion covers a large number of proteins including the FtsWI complex (sPG synthesis machinery), FtsBLQ (regulator), FtsN (trigger), and FtsA (FtsN recruiter). Structural models are employed to show the position of mutations identified within these proteins, which the authors use to produce successive models for septal peptidoglycan synthesis. The structures shown in Figures 9 and 13 were especially helpful in understanding how FtsL may regulate septal peptidoglycan synthesis. Overall, the authors do an expert job in distilling a large and admittedly complicated set of literature into manageable vignettes that should be helpful to those interested in cell division and cell wall synthesis. I have a few comments, all of which are minor.

Comments

Lines 48-49: “However, historically speaking compounds directly targeting peptidoglycan biosynthesis 48

have been one of the most effective classes of antibiotics [11].” The authors should state a couple of examples of these compounds here (e.g., beta-lactams, vancomycin, etc.).

Line 63: “FtsN is another regulatory protein, which arrival…” to “whose arrival…”

Lines 96-97: “One of the most significant ones is the observation of an independently of FtsZ-ring dynamics slowly moving PG synthesis track.” This statement is awkward.

Lines 103-104: “FtsEX also activates the amidase activity of AmiA and AmiB…” The authors should be more specific here. FtsEX activates AmiA and AmiB through EnvC.

Lines 114-115: “DedD is the only other protein next to FtsN that contains a PG-binding SPOR domain and results in division defects when absent.” The authors should be more specific here since mutants lacking DamX and DedD grow long (PMID: 19684127 and 19880599) and mutants lacking RlpA chain in Pseudomonas aeruginosa (PMID: 24806796) and Vibrio cholerae (PMID: 31286580). See also lines 514-515.

Line 128: (C55-P-NAM-NAG) to (C55-PP-NAM-NAG)

Lines 141-143: “PG hydrolases are reported to drive the separation of the daughter cells and the amidases in particular are indicated to be involved in the spatial distribution of the sPG synthesis complex [8,44].” The reader would be helped if the authors also noted that lytic transglycosylases are required to separate daughter cells. These include mutants lacking one (PMID: 24806796), two (31286580), or three (PMID: 12399477) lytic transglycocylases.

Lines 160-161:  Lipid II is then flipped to the periplasmic side of the inner membrane by a flippase (most likely Mur)…” Is there a reason why the authors are not confident that MurJ is the lipid II flippase? The authors may also consider including Amj (PMID: 25918422) for completeness.

Line 165: “peptidases” to “Peptidases”

PBP1b as a repair enzyme

Lines 199-202, 494-497, and elsewhere: The authors introduce the reader to the possibility that PBP1b is a repair enzyme. If that is the case, could the authors comment on why they think a repair enzyme is activated by FtsN at the start of constriction? Do the authors think that FtsWI are prone to error? Is PBP1b required to prime septal peptidoglycan synthesis?

Line 538: “[28,29]..” to “[28,29].”

Line 814: “leads do a weak dominant” to “leads to a weak dominant”

Line 964: “Figure 15” is in a different font from the text

Line 999: “PPB1” to “PBP1b”

Figure 15: Could the authors label FtsI in all panels? FtsN should also be labeled. Also, the authors should consider increasing the font size of the protein names.

Line 1040: “most likely direct” to “most likely directs”

Lines 1058-1060: “After sufficient endopeptidase and amidase activity and subsequent FtsN accumulation, the number of synthesis complexes translocated to the slower moving synthesis track away from the FtsZ assembly track increase (Figure 8).” This statement is confusing to read.

Lines 1086-1087: “PG strands are likely shorter at the poles than at cylindrical part of the cell…” Could the authors explain why they think PG strands are shorter at the poles?

Reviewer 2 Report

This review summarized an updated model of the divisome complex, especially based on structural, biochemical, and genetic data. The authors analyzed the functions of diverse divisome components, including FtsWI, PBP1b, FtsA, FtsN, DedD, FtsBLQ, FtsEX, and peptidoglycan amidases. Particularly, recent results about the roles of FtsN and FtsBLQ were extensively analyzed. Based on these data, the authors suggested a speculative model of peptidoglycan synthesis at septum. Additionally, the article presented the three-dimensional view of an architecture model of the divisome complex.

1. Please add a novel figure describing the recruitment order of the divisome complex, in order to help readers understand better.

2. L96-97; an independence

Author Response

see attached document
